

# Investigating long-term changes in polar stratospheric clouds above Antarctica: A temperature-based approach using spaceborne lidar detections

Mathilde Leroux[1] and Vincent Noel[1]

[1]LAERO, Laboratoire d'Aérologie, Université Toulouse III Paul Sabatier, CNRS, Toulouse, 31400, France

*Correspondence to*: Mathilde Leroux (mathilde.leroux@aero.obs-mip.fr)

**Abstract.** Polar stratospheric clouds play a significant role in the seasonal thinning of the ozone layer by facilitating the activation of stable chlorine and bromine reservoirs into reactive radicals, as well as prolonging the ozone depletion by removing $HNO_3$ and $H_2O$ of the stratosphere by sedimentation. In a context of climate change, the cooling of the lower polar stratosphere could enhance the PSC formation and by consequence cause more ozone depletion. There is thus a need to document the evolution of the PSC cover to better understand its impact on the ozone layer. In this article we present a statistical model based on the analysis of the CALIPSO PSC product from 2006 to 2020. The model predicts the daily regionally-averaged PSC density by pressure level derived from stratospheric temperatures. Applying our model to stratospheric temperatures from the CALIPSO PSC product over the 2006-2020 period shows it is robust in the stratosphere between 10 and 150 hPa, reproducing well PSC variations over daily timescales and seasonal differences (2006-2020). The model reproduces well the PSC seasonal progression, even during disruptive events like stratospheric sudden warmings, except for years characterized by volcanic eruptions. We apply our model to gridded stratospheric temperatures from reanalyses over the complete south pole domain to evaluate changes in PSC seasons over the 1980-2021 period. We find two distinct periods in the evolution of the PSC season duration. Between 1980 and 2000, the PSC season increased by 15 days at 10-20 hPa with an increasing lengthening as we descend in altitudes to reach 30 days at 100-150 hPa. This lengthening is in possible relation with major volcanic eruptions occurring over this period. After 2000, a temporary drop mostly visible at high (10-20 hPa) and lower altitude (100-150 hPa) is followed by a progressive increase in PSC season duration. Over the 1980-2020 period, the PSC season increased by 20 days between 30-100 hPa. These changes are altitude-dependent and statistically significant. We discuss the impact of non-temperature stratospheric changes on the variations of PSC seasons.





# 1 Introduction

In the 1980s, the attention of the scientific community was captured by the discovery of a hole in the ozone layer. Research revealed that ozone thinning is driven by the presence of chlorine (Molina and Rowland, 1974) that is mainly emitted by human activities in the form of chlorofluorocarbons (CFCs), which persist in the atmosphere for a long time. Photolysis, a chemical

reaction triggered by sunlight, converts the CFCs into active chlorine, which then reacts with ozone, leading to its depletion. However, the high levels of chlorine observed at the poles could not be entirely explained by the presence of CFCs alone (Solomon et al., 1986). Polar Stratospheric Clouds (PSCs) emerged as a significant contributor to this phenomenon.

PSCs form in the polar stratosphere between 15-30 km in altitude, during hemispheric winters when the polar vortex lowers

the stratospheric temperature sufficiently (T < 195 K) to enable their formation. Cold temperatures trigger chemical nucleation processes involving water vapor ($H_2O$), nitric acid ($HNO_3$) and acid sulfuric ($H_2SO_4$), causing the formation of three types of particles that are often mixed within PSCs. 1) Liquid droplets of supercooled ternary solution (STS) form when water vapor and nitric acid condense on stratospheric background aerosols. 2) Solid particles of nitric acid trihydrate (NAT) exclusively form through heterogeneous nucleation processes, occurring either on ice particles or meteoritic nuclei (e.g. Hoyle et al., 2013,

James et al., 2018). The nucleation of NAT particles on meteoritic nuclei was first documented by laboratory studies (e.g. Bogdan et al. 1999, 2003) and then highlighted by observations which showed NAT PSC formation despite an absence of stratospheric dehydration and temperatures above the freezing point. This process leads to the formation of low-density, large NAT particles, known as "NAT-rock" or "Mother-NAT" (Tritscher et al., 2021). 3) Ice crystals also form in the polar stratosphere, through both homogeneous and heterogeneous nucleation. Homogeneous nucleation is a well understood

pathway of formation which requires extremely low temperatures (3-4 K below the freezing point). Such low temperatures can be generated by gravity waves triggered by orography, and lead to the fast formation of a specific PSC type known as wave ice PSC. NAT particles can then nucleate on these ice particles and propagate over vast regions, a phenomenon identified through spaceborne measurements and called the "NAT-belt" (e.g. Höpfner et al., 2006, Noel and Pitts, 2012). Heterogeneous nucleation of ice PSC particles can occur over pre-existing NAT particles (Fortin et al., 2003) or foreign nuclei. Even though

temperatures are key in driving the formation of specific PSC particles, other parameters are important such as water vapor. The formation of PSCs is very sensitive to changes in water vapor: studies had shown that a wetter and cooler stratosphere would produce more PSC (e.g. Stenke et al., 2005, Khosrawi et al., 2016). Measurements taken at a single mid-latitude location (Boulder, Colorado) show a 25 % net increase in stratospheric water vapor since 1980. However, when considering merged satellite data records since the late 1980s, there appears to be little overall net change, which suggests that the trend observed

in Boulder is not representative of the global stratosphere (Hegglin et al., 2014). According to the AR6 report there is a low confidence in trends of stratospheric water vapor over the instrumental period (AR6, IPCC, chap.2, 2021).



PSCs have a dual role in the thinning of the ozone layer. First, by being the site of heterogeneous reactions, they activate the stable reservoirs of chlorine and bromine of the stratosphere into free radicals which subsequently interact with ozone and contribute to its destruction during spring with the return of sunlight (Solomon, 1999). Secondly, they induce denitrification and dehydration of the stratosphere by capturing the $HNO_3$ and $H_2O$ necessary to their formation, which delays the deactivation of chlorine and thus prolongs the ozone depletion (e.g. Toon et al., 1986; Jensen et al., 2002, Khosrawi et al., 2017). Although since the implementation of the Montreal Protocol ozone-depleting substances like CFCs have decreased (Montzka et al., 2003), fluctuations in stratospheric temperatures also influence the future recovery of stratospheric ozone. Indeed, the rate of chemical reactions causing ozone destruction and PSCs formation are temperature-dependent (Eyring et al., 2007). Moreover, the increasing trend in greenhouse gasses (GHG) due to anthropic emissions induces a cooling effect of the lower stratospheric levels (AR6, IPCC, 2021) which could enhance PSC formation, resulting in increasing ozone depletion despite a decrease in stratospheric chlorine loading (e.g. Shindell et al., 1998, Schnadt et al., 2002). Equilibrium-doubled $CO_2$ experiments in different models and experimental setups confirmed this trend by predicting a cooler stratosphere (Wang et al., 2020). It is therefore crucial to document the PSC cover and understand what drives its evolution in a context of climate change to better understand its impact on the reformation of the ozone hole.

Our goal is to better understand if changes in PSC cloud cover over climate timescales could potentially affect the reformation of polar stratospheric ozone. To make progress toward this goal, our objective in this paper is to evaluate if the seasonal evolution of PSC over Antarctica has gone through significant changes over the past decades (1980-2021). We first use PSC detections obtained from spaceborne lidar measurements between 60-82°S over the 2006-2020 period to develop a statistical model that predicts by pressure level the daily average PSC density over the polar region based on stratospheric temperatures. We then apply this model to a gridded dataset of stratospheric temperatures from reanalyses that cover the entire Antarctica region (60°S-90°S) to evaluate seasonal PSC changes over the past decades.

We first present the data used for our study in Section 2, including observations and reanalysis. Section 3 outlines the methodology to create our statistical model and section 4 documents its performance over two extreme PSC seasons. In Section 5, we present PSC time series spanning the period from 2006 to 2020 to verify the robustness of our model. We compare the observed evolution of PSC density, as measured by spaceborne lidar, with the simulated PSC density generated by our model. In section 6, we apply our model to the gridded stratospheric temperatures from reanalysis covering the years 1980 to 2021, allowing us to document the evolution of PSC seasons. In section 7, we discuss the influence of stratospheric sudden warmings and volcanic eruptions on our model predictions. Section 8 consists of a discussion and conclusion based on the results obtained in the study.



## 2 Data

### 2.1 Spaceborne lidar PSC product

From 2006 to 2018, CALIPSO (Cloud-Aerosol Lidar and Infrared Pathfinder Satellite Observations) was part of the A-Train, a constellation of satellites flying at an altitude of 705 km above the sea level (ASL) along a sun-synchronous polar orbit inclined at 98° (Winker et al., 2009). CALIPSO crossed 15 times the equator per day and provided a coverage between 82°N and 82°S, making it a satellite of interest for the study of the polar regions. In 2018, both CALIPSO and the radar-based

CloudSat moved to a lower orbit called the C-Train, 16.5 km lower (Braun et al., 2019). CALIPSO ended operations in July 2023.

CALIOP (Cloud-Aerosol LIdar with Orthogonal Polarization) is the main instrument of the CALIPSO satellite. CALIOP is a polarized backscatter lidar at two wavelengths (532 and 1064 nm) which measures backscatter from atmospheric components.

The lidar shoots a laser pulse every 335 m along the ground track at 3° off nadir. The return signal is recovered by a 1-m telescope whose viewing footprint diameter is 90 m at the ground. CALIOP is very sensitive to vertical variations of atmospheric components which let us determine with precision the presence and altitude of layers of clouds or aerosols. Moreover, lidar observations have a good sensitivity to optically thin layers, allowing easy detection of PSC. The nominal vertical resolution of a CALIOP profile begins at 30 m in the lower troposphere, and increases to 180 m from 20.2 km to 30.1

km, then 300 m beyond to 40 km ASL.

The CALIOP level 1b product provides three types of geolocated profiles: the total backscatter coefficient at 532 and 1064 nm and the perpendicular backscatter coefficient at 532 nm (Winker et al., 2009). These profiles are used to create cloud and aerosols products (e.g. Chepfer et al., 2013, Liu et al., 2019). Here we use the PSC product level 2 (LID_L2_PSCMask-

Standard-V2-00), obtained by applying the V2 PSC detection and composition classification algorithm to CALIOP level 1B data as described in Pitts et al., 2009 and 2018. PSC are identified when the perpendicular backscatter coefficient and the backscatter ratio at 532 nm exceed the background levels corresponding to clear sky conditions. The perpendicular backscatter coefficient is very sensitive to non-spherical solid particles, allowing the identification of PSCs NAT and ICE. Unlike STS particles, which consist of liquid particles and do not cause depolarization, both NAT and ice particles are solid and induce

depolarization. To differentiate between NAT and ice, the focus is on backscatter intensity; a significant increase in backscatter indicates the presence of ice.

The CALIPSO PSC product describes the spatial distribution, composition, and optical properties of PSC layers along CALIPSO orbit tracks. The product is presented as daily files. Each daily file contains one vertical profile every 5 km along-

track, containing 121 altitude levels between 8-30 km ASL with a vertical resolution of 180 m. From May to October daily



files contain profiles sampled over the South pole region (50-82°S), from December to March daily files contain profiles sampled over the North pole region (50-82°N). Those periods correspond to hemispheric winters. Here, we focus only on the South pole (May-October). The product includes complementary information, such as the stratospheric temperature from reanalysis (Sect. 2.2) interpolated on the PSC data product grid, and mixing ratios for species essential to PSC formation such

as $HNO_3$ or $H_2O$ derived from Microwave Limb Sounder (MLS) measurements.

In the CALIPSO PSC product, PSC detection is limited to nighttime observations as they ensure a high signal-to-noise ratio. This means CALIPSO sampling changes along a PSC season (Appendix A). Near the end of the season, in particular, enhanced sunlight induces poor sampling. In October, a large area of the polar region near 90°S is being illuminated by the Sun, and

most of the nighttime measurements are spread between 55°S-75°S (Pitts et al., 2007). Moreover, CALIPSO sampling changes across latitude bands. Near 82°S, the density of CALIPSO profiles increases due to intersections of CALIPSO overpasses, while further away from the pole, the number of profiles per latitude band decreases. Those elements are taken into account in our analysis by ignoring days with limited sampling and by weighting zonally our average PSC densities above Antarctica. In the rest of the study, when the total number of points sampled by CALIPSO for a given day falls below one third of the

maximum number of points over a season, the day is not considered.

## 2.2 Globally-gridded temperatures from MERRA2 reanalysis

From the CALIPSO PSC product, we use stratospheric temperatures from the Modern Era Retrospective analysis for Research and Application (MERRA2) which are produced by the NASA Global Modeling and Assimilation Office (GMAO). MERRA2 incorporates aerosol data assimilation and ozone representation. In the CALIPSO PSC product, these stratospheric

temperatures are interpolated to the PSC data product grid, into profiles and altitude levels.

In addition to the stratospheric temperatures contained in the CALIPSO PSC product, we consider stratospheric temperatures from the gridded MERRA2 dataset. We utilize the M2I6NPANA.5.12.4 collection, which consists of analyzed meteorological fields including temperature, and is available at a 6-hour time frequency starting from 00:00 UTC daily from 1980 to the

present (Gelaro et al., 2017). The spatial grid has a horizontal resolution of 0.5° x 0.625° (Latitudes x Longitudes). The vertical structure of this collection is configured with pressure levels, encompassing 42 levels ranging from 1000 to 0.1 hPa, with 9 levels allocated in the stratosphere: 10, 20, 30, 40, 50, 70, 100, 150, and 200 hPa.

## 3 Temperature-based model of PSC density

This section provides an overview of the methodology followed to construct our statistical model. Our intent for our model is

to predict the daily average PSC density in the polar region, at a given pressure level, based on geographical maps of





stratospheric temperature. In the first part, we describe the successive steps necessary to build our model which consists of combining PSC detections from the CALIPSO PSC product with temperatures from reanalysis above the South pole. In the second part, we explain how for each pressure level, we choose the temperature threshold required to connect PSC densities with temperature maps. Finally, we explain how our model can be applied to stratospheric temperatures to predict PSC cover.

## 3.1 Statistical model design

We construct a 3d spatial grid (Latitude x Longitude x Pressure) above Antarctica from 82°S to 60°S. Each gridbox is 2°x4°. This resolution provides each gridbox with a reasonable number of CALIOP samples on a daily scale, and is close to the horizontal resolutions of General Circulation Models (GCMs). Vertically we use the pressure levels of the gridded MERRA2 data (Sect. 2.2) to make our model relevant to reanalysis resolutions.

In CALIPSO PSC daily files (Sect. 2.1), variables are organized into profiles containing 121 altitude levels from 8 to 30 km. A mask variable indicates if there is a PSC or not at each vertical level of every profile, and if the PSC location is close or far above the tropopause. In each profile we use the mask variable to identify points with a PSC just above the tropopause. In the following, we make no distinction of PSC type (STS, Enhanced NAT, Liquid NAT, ICE and Wave ICE) and consider them all together.

For each day, we extract from the PSC daily file, for a pressure level $P$, in each lat-lon gridbox, the number of sampled points $N$ (Fig. 1a), the number of PSC detections $n$ out of those sampled points (Fig. 1b), and calculate in each gridbox PSC densities as the ratio $F(P) = \frac{n}{N}$ (Fig. 1c). We also calculate the average stratospheric temperature $\bar{T}$ from MERRA2 over the sampled points (Fig. 1d). For 2009/07/01, at 10-20 hPa, the PSC density is higher in the southeast region from 82°S to 70°S and 0°E to 120°E (Fig. 1c), where cold temperatures reach below 190K (Fig. 1d). Conversely, warmer temperatures lead to smaller PSC densities, for instance 60°W-120°W.



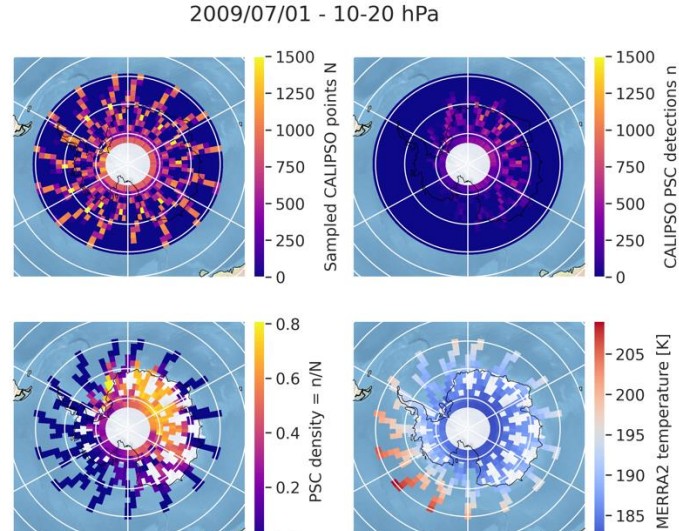

**Figure 1: Output of daily gridding for 2009/07/01, between 10-20 hPa. (a) number of sampled points $N$. (b) number of PSC detections $n$. (c) PSC density $F(P)$, and (d) mean stratospheric temperature $\overline{T}$ from MERRA2.**

For each pressure level $P$, we spatially average daily gridded maps of PSC density $F(P)$ (Fig. 1c) over the region to obtain a mean daily PSC density $\overline{F(P)} = \frac{\sum_{lat,lon} F(P)*w}{Nb}$, with $Nb$ the number of lat-lon gridboxes sampled by CALIPSO and $w$ the weight as a function of latitude to take into account the zonal variation in gridbox size. In parallel, we use the map of mean stratospheric temperature $\overline{T}$ (Fig. 1d) to create for each pressure level $P$ a map of cold density. To do this, we establish a temperature threshold $T_{psc}$ (the way $T_{psc}$ is selected is described in Sect. 3.2), and in each lat-lon gridbox we check if the mean stratospheric temperature $\overline{T}$ is colder than this threshold. We obtain a map filled with 1 and 0 depending on whether the condition is met or not. We then calculate the daily cold density $\overline{C(P)} = \frac{\sum_{lat,lon} x*w}{Nb}$ with x = 1 when ($\overline{T} < T_{psc}$), $w$ the weight as a function of latitude and $Nb$ the number of lat-lon gridboxes sampled by CALIPSO. The daily $\overline{F(P)}$ and $\overline{C(P)}$ sum up the maps of Fig. 1.c and 1.d as a couple of single numbers for the entire polar region at a pressure level $P$. These values provide a convenient way to describe the extent of the PSC spatial cover on a given day, enabling the study of PSC evolution over long periods.

We followed the steps above to get ($\overline{F(P)}, \overline{C(P)}$) for each pressure level $P$ and day of the PSC dataset, and established a relationship between these variables for each pressure level and month of the PSC season (May to October). Below we show an example of this process considering the 2009 winter. Figure 2 shows $\overline{F(P)}$ and $\overline{C(P)}$ for the month of July, 2009 at three pressure levels. One point in these plots sums up a daily map like in Figure 1. There are 31 points in each scatterplot, corresponding to the number of days in July. The regression that fits the best for most of the plots is a polynomial of degree 2





(red lines, Fig. 2). The regression equations take the form: $\overline{F(P)} = a * \overline{C(P)}^2 + b * \overline{C(P)} + c$. These equations relate $\overline{F(P)}$

and $\overline{C(P)}$, for each pressure level $P$ during the month of July, 2009, and constitute our PSC prediction model. They allow us to produce a mean simulated PSC density called $\overline{F'(P)}$ from a known mean cold density $\overline{C(P)}$. This methodology is repeated for each month of the 2009 season. In the next subsection, we explain the process of selecting $T_{psc}$ and its significance in our analysis.

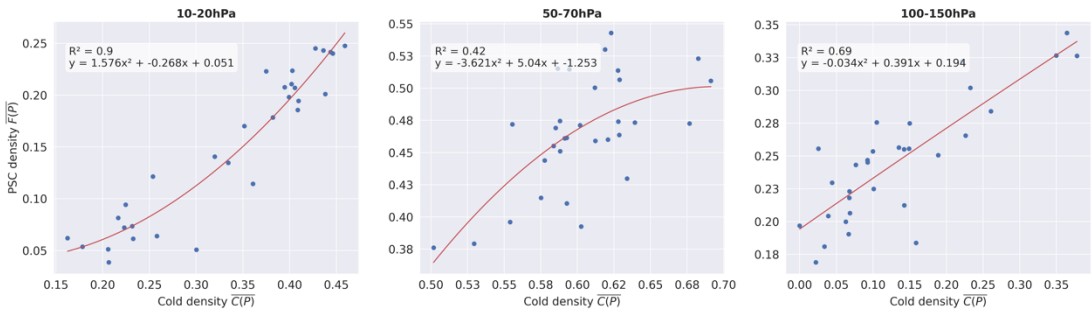


**Figure 2: Daily PSC densities $\overline{F(P)}$ vs cold densities $\overline{C(P)}$ for July 2009 (31 points in each panel). The red line is the trend line with its equation in the left white panel. Panel (a): 10-20 hPa ($T_{psc}$ = 187 K). Panel (b): 50-70 hPa ($T_{psc}$ = 192 K). Panel (c): 100-150 hPa ($T_{psc}$ = 194 K).**

## 3.2 Determination of $T_{psc}$

The selection of the temperature threshold $T_{psc}$ is a crucial step. It is the temperature above which PSC are not detected in the CALIPSO PSC Product, and above which our model will not predict PSC presence in a gridbox. Here, we select $T_{psc}$ as the temperature threshold that optimizes the agreement between the mean PSC density $\overline{F(P)}$ observed by CALIOP and the simulated mean PSC density $\overline{F'(P)}$ generated by our model over the entire time period considered. In addition to cold temperatures, PSC formation requires appropriate concentrations of chemical species like HNO$_3$ and H$_2$0. These concentrations are not constant temporally or spatially, which is why we calculate a different $Tpsc$ for each pressure level and each month.

$T_{psc}$ can be calculated over a specific winter (Sect. 4) or longer periods such as multiple PSC seasons (Sect. 5).

To find the optimal $T_{psc}$, for each pressure level P and each month, we consider a wide range of candidate temperature thresholds $T_{psc}$ from 187 K to 207 K. For each candidate $T_{psc}$, we calculate the mean cold density $\overline{C(P)}$ and derive regression

parameters between $\overline{C(P)}$ and $\overline{F(P)}$ (Sect. 3.1). Based on these parameters we calculate the daily average PSC densities $\overline{F'(P)}$ and calculate the mean absolute error (MAE) between $\overline{F'(P)}$ and $\overline{F(P)}$ over the month, which quantifies the discrepancy





between the observed and simulated data for given regression parameters. The $T_{psc}$ and parameters which lead to the smallest MAE are selected for the month and pressure level considered.

Below we compare the retrieved $T_{psc}$ with the temperature of nitric acid nucleation $T_{NAT}$, which is the warmest temperature at which PSC particles can in theory start to form and subsist, considering all available chemical species. We calculate $T_{NAT}$ from the profiles of stratospheric temperature and of mixing ratio for $HNO_3$ and $H_2O$ reported in the PSC product (Sect. 2.1). Those values are vertically distributed over specific pressure levels, called $P(HNO_3)$ and $P(H_2O)$. For each daily file in the PSC product, we extracted $HNO_3$ and $H_2O$ mixing ratios between 10 and 250 hPa, and reinterpolated the $H_2O$ mixing ratios on

$HNO_3$ pressure levels. Using the formula from Hanson and Mauersberger (1988), we calculated daily $T_{NAT}$ from the mixing ratios and temperature at each pressure level. We interpolated $T_{NAT}$ on the 121 altitude levels of the CALIPSO PSC product (Sect. 2.1). As in Sect. 3.1, we averaged the $T_{NAT}$ in our 3d spatial grid (Latitude x Longitude x Pressure), and calculated the latitude-weighted spatial average to obtain a mean daily $T_{NAT}$ vertical profile. Finally, we averaged these daily profiles to get an average $T_{NAT}$ vertical profile for a given time period.


The $T_{NAT}$ vertical profile we calculated over the 2006-2020 period (Fig. 3) is consistent with the literature: at 14 km altitude (100 hPa), typical stratospheric mixing ratios are around 3 ppm for $H_2O$ and 5 ppb for $HNO_3$, resulting in a $T_{NAT}$ value of about 197 K (Hanson and Mauersberger, 1988), similar to our results. At 50 hPa, mixing ratios of 10 ppbv for $HNO_3$ and 5 ppmv for $H_2O$ lead to a $T_{NAT}$ value of about 195.7 K (Pitts et al. 2018), consistent with our result of 194 K. The $T_{psc}$ vertical

profile we calculated over July over the 2006-2020 period following the methodology above are relatively close to $T_{NAT}$ profiles, especially in July. Both temperature thresholds become colder in the upper stratosphere and warmer at lower altitudes, except at 10-20 hPa for the September and October $T_{psc}$ profiles. Apart from July, $T_{psc}$ are generally significantly colder than $T_{NAT}$ (more than several standard deviations apart). The difference is particularly important at lower altitudes (30-100 hPa).



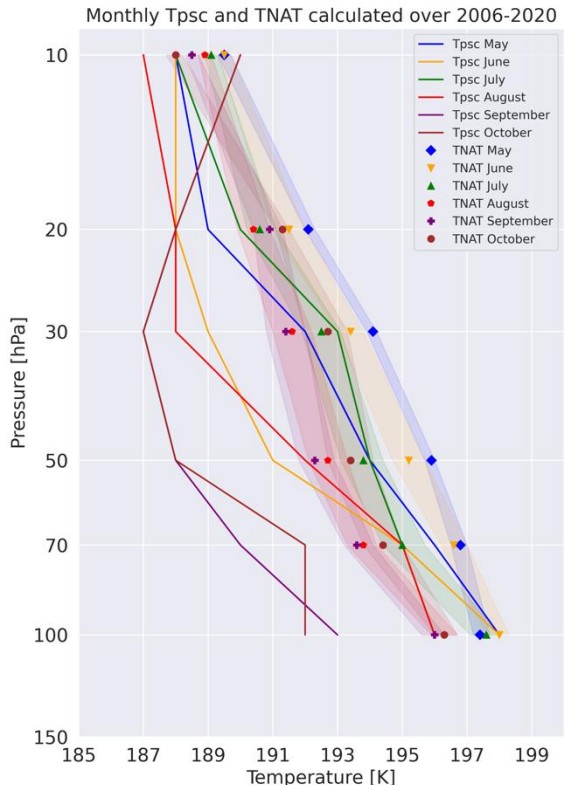

**Figure 3: Vertical profiles of monthly $T_{psc}$ (colored lines) and $T_{NAT}$ (colored symbols). The colored envelopes represent the standard deviation of daily $T_{NAT}$.**

PSC particles observed by CALIPSO in the lower stratosphere can come from nucleation processes when temperatures are below the threshold formation temperature $T_{NAT}$, but they also can be sedimenting from upper levels. The formation of PSC particles requires a long time in cold temperatures, thus the mere occurrence of temperatures colder than $T_{NAT}$ might not be sufficient for PSC detection in CALIPSO measurements. NAT particles may form but in too small concentrations to be detected. All these elements could explain why there is often a large difference between the temperature threshold $T_{NAT}$ at which PSC can form and the temperature threshold $T_{psc}$ at which PSC are observed. In the end, using $T_{psc}$ significantly colder than $T_{NAT}$ (as in Fig. 3) provides the best match between observed and predicted PSC densities.

## 3.2 Simulating PSC densities $\overline{F'(P)}$

Once we have derived appropriate regression parameters (Sect. 3.1) and $Tpsc$ (Sect. 3.2) for each month and pressure level, we can use those to generate PSC densities $\overline{F'(P)}$ on daily timescales across a specific period. To do so, for a given pressure level, we require a daily gridded map of temperature at large-scale spatial scales. This map can be either obtained by regridding





255 the temperatures profiles from the CALIPSO PSC product (Sect. 2.1), or straight from the MERRA2 reanalysis dataset (Sect. 2.2). Considering the appropriate $T_{psc}$ for the pressure level under study, from the daily map of temperatures we derive the daily average cold density $\overline{C(P)}$. We then apply the polynomial model (as in Sect. 3.1, Fig. 2) to the daily average cold density $\overline{C(P)}$, to generate as output a daily average simulated PSC density $\overline{F'(P)}$.

## 4 A close look at the 2009 and 2010 Antarctic winters

260 ### 4.1 The 2009 Antarctic winter

According to the bulletin from the World Meteorological Organization (WMO), 2009 was characterized by a stable and elongated vortex, resulting in a significant expansion of the PSC cover (particularly NAT), causing early ozone depletion. Figure 4 illustrates the daily variations in PSC densities from May 1$^{st}$ to October 31$^{st}$ 2009, observed by CALIPSO ($\overline{F(P)}$, Sect. 2, full lines) and simulated by our model ($\overline{F'(P)}$, Sect. 3.3, red dots). The monthly temperature thresholds $T_{psc}$ used for

265 this season are reported in the table 1 of Appendix B.



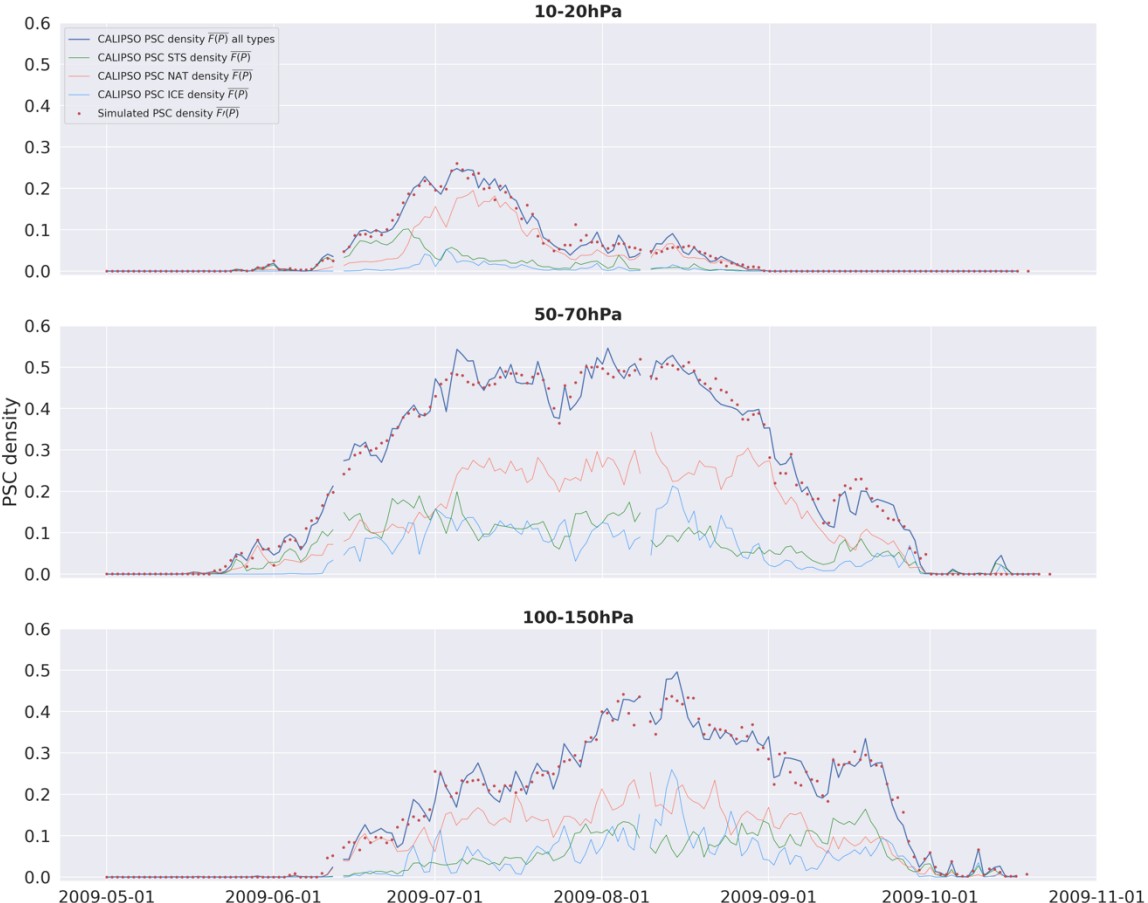

**Figure 4: Time series of daily average PSC densities $\overline{F(P)}$ observed by CALIPSO (blue curve) and simulated by our model $\overline{F'(P)}$ (red dots) for the 2009 season on pressure levels (a) 10-20 hPa, (b) 50-70 hPa and (c) 100-150 hPa. Colored lines represent the different PSC types: Green for STS, orange for NAT and blue for ICE. Interruptions in the blue curve are indicative of days with insufficient CALIPSO sampling (Sect. 2.1).**

At 10-20 hPa (Fig. 4, top), observed PSC densities begin to increase in mid-June. PSC densities are highest (> 0.2) between June 28[th] and July 14[th], i.e., early July. PSC densities fall below 10% around July 20[th], and remain stable until the end of August. Densities at this pressure are low, and the PSC density exceeds 10% during one month only. At 50-70 hPa (Fig. 4b), PSC are observed by CALIPSO more frequently, over longer periods. They appear at the end of May and persist until the end of September. The limited nighttime sampling beyond that date (Sect. 2.1) prevents us from knowing the exact season end date. PSC density is maximum in July-August, reaching values above 50%. At 100-150 hPa, observed PSC densities increase from mid-June to early August, when they reach their peak (>0.4) for a few days, and then gradually decrease until October 1[st], with a sharp drop the last week of September. At this pressure level, PSCs appear later in the season because temperature





changes in the vortex appear first at high altitudes and then propagate downward. Moreover, daily variations in PSC densities are more important.

Simulated PSC densities ($\overline{F'(P)}$, red dots in Fig. 4) generally agree very well with observed PSC densities ($\overline{F(P)}$, blue). At all
pressure levels, the difference between $\overline{F'(P)}$ and $\overline{F(P)}$ is negligible (Appendix B, Fig. B1), never exceeding 0.075. Our model generally performs better at higher altitude where the PSC cover and temperatures are more stable, but reproduces well the global shape of the PSC seasonal evolution even at lower altitudes. PSC NAT density falls faster than density of other PSC types at 50-70 hPa during September, going from 0.3 to 0.1 in 2 weeks (Fig. 4b, orange curve). The sedimentation of the NAT particles generates a decrease in $HNO_3$ concentrations, adding a constraint to PSC formation. Whereas until now temperature
was the main constraint on PSC formation, now it is the availability of $HNO_3$. $Tpsc$ at 50-70 hPa and 100-150 hPa become colder at the end of the season (as in Fig. 3), a symptom of the denitrification. Overall, our model performs very well in predicting the evolution of PSC coverage during the 2009 winter with particularly cold stratospheric temperatures extended over a long time period.

## 4.2 The 2010 Antarctic winter

In July 2010, the Antarctic polar vortex was disturbed by strong dynamic activity. A negative quasi biennial oscillation (QBO) had favored the channeling of planetary waves, which produced a warming at high altitudes and a weakening of the polar vortex, especially at 10 hPa (Klekociuk et al., 2011). The warming was intense during July, with a peak temperature rise of over 20 K on July 31[st] at 10 hPa, and persisted into August. This led to one of the smallest ozone holes in the 15 years prior (WMO, 2010). The monthly temperature thresholds $T_{psc}$ used for this season are reported in the table 2 of Appendix C. Fig. 5
reports the observed and predicted PSC densities for the 2010 winter.





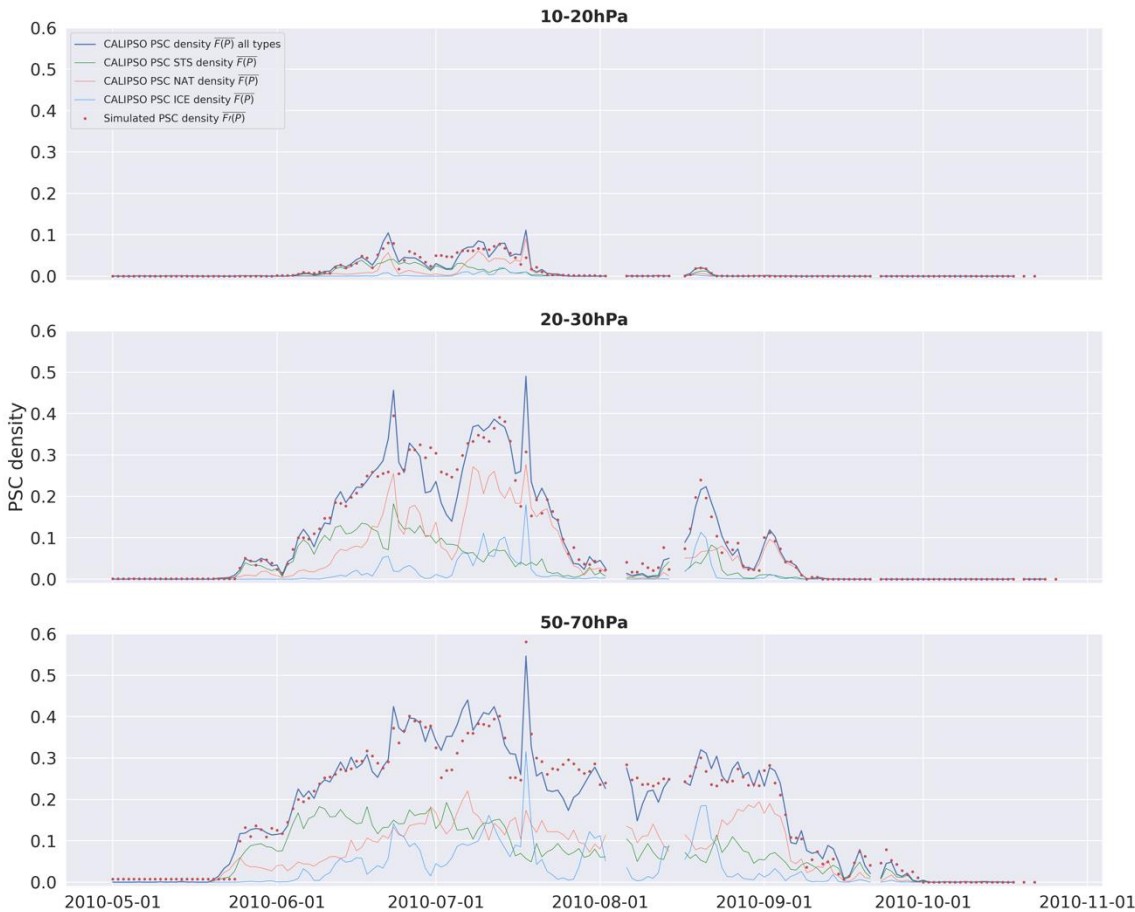

**Figure 5: Same as Figure 4, for the 2010 PSC season, on pressure levels (a): 10-20 hPa, (b): 20-30 hPa and (c): 50-70 hPa.**

At 10-20 hPa (Fig. 5a, top), very low PSC densities are observed, never exceeding 10%. A minor warming appeared from June
20th to June 30th and a major one from July 14th to August 1st (Appendix C, Fig. C1). These two warming events correspond
with two drops in PSC densities visible at 10-20 hPa but also at 20-30 hPa (Fig. 5b). At 50-70 hPa, temperatures appear less
affected and July PSC densities are quite high with values above 40%. PSC densities, however, decrease from mid-July to
early-August in the warming period (Fig. 5c). Observations show PSCs start to reform at 20-30 hPa from mid-August to early-

September, indicating the end of the warming (Fig. 5b). At all pressure level, the observed PSC densities peak on June 23rd
and July 18th, and are smaller in between. At 10-20 hPa and 20-30 hPa, those peaks are made of NAT (orange) while at 50-70
hPa, they are made of ice particles (blue), suggesting a more intense cooling at lower altitudes. The daily evolution of MERRA2
temperatures (not shown) suggests these peaks correspond to rapid cooling at sub-daily scale. PSC densities follow temperature
fluctuations very closely, meaning that PSCs form and dissipate in response to cooling or warming on sub-daily time scales.

Our model is therefore appropriate for representing PSC variations due to temperature changes on short time scales.





PSC densities simulated from our model (red) are generally very close to observed PSC densities (blue), even under intense temperature variations on short timescales. In the presence of sudden peaks, our model reproduces PSC densities but slightly underestimates their maximas. Model performances are best at high altitude with extremely low average and standard deviation

of differences between $\overline{F'(P)}$ and $\overline{F(P)}$ (Appendix C, Fig. C2). At the 50-70 hPa pressure level, maximum differences between simulated and observed PSC densities can reach 0.2, for instance in July for a PSC peak.

In this section, we saw that our model is able to reproduce the daily variations of observed PSC densities over single seasons with a high level of accuracy, even in presence of strong and rapid temperature variations. In the next section, we will apply

our methodology to the CALIOP observational period (2006-2020) and discuss results of our model over longer timeframes.

## 5 Evolution of PSC densities over 14 years (2006-2020)

In this section, we compare the daily evolution of the observed PSC density $\overline{F(P)}$ with the simulated PSC density $\overline{F'(P)}$ generated by our model applied to MERRA2 temperatures contained in the PSC product over the 2006-2020 period. We follow the same method as for the 2009 and 2010 single seasons (Sect. 4). The temperature threshold $T_{psc}$, required to obtain $\overline{C(P)}$

(Sect. 3), is calculated for each pressure level from 2006 to 2020, by aggregating daily values from similar months together. Examples of regression performance are shown in Appendix D. Here we focus our analysis on three pressure levels: 10-20 hPa (Sect. 5.1), 50-70 hPa (Sect. 5.2) and 100-150 hPa (Sect. 5.3).

### 5.1 10-20 hPa pressure level

Figure 6 (top) depicts the observed daily PSC densities $\overline{F(P)}$ (Sect. 2.2) from 2006 to 2020 at 10-20 hPa (blue lines). Observed

densities are low compared to other pressure levels. In 2010 and 2012, observed PSC densities are particularly low, with a maximum never exceeding 10%, due to a warm vortex (Pitts et al., 2018). On average over the 2006-2020 period, observed densities (Fig. 6b, blue) start to increase in early June and vanish late August. Observed PSC densities reach on average 12% of the Antarctic surface around on July 13[th], when the vortex is fully extended. This coincides with the date on which MERRA2 temperatures from the PSC product, daily averaged over 60°-82°S, then seasonally averaged over the 2006-2020 period (not

shown), reach their minimum (July 10[th]). This suggests that the stratosphere at 10-20 hPa reaches its cold peak in an environment where the chemical species necessary for the formation of PSCs are still abundant.



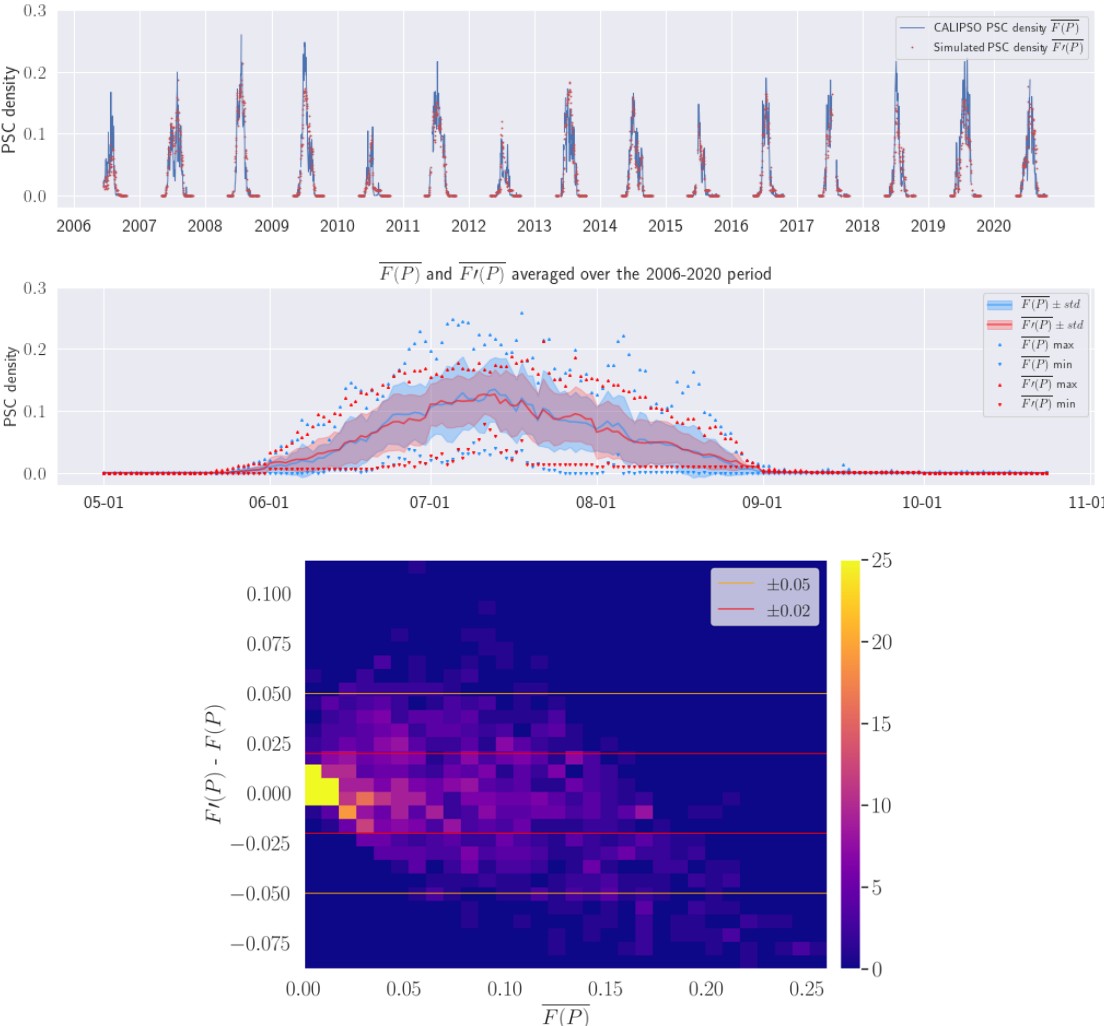

**Figure 6: (a) Time series of daily average PSC densities observed by CALIPSO $\overline{F(P)}$ (blue) and simulated by our model $\overline{F'(P)}$ (red) from 2006 to 2020 for the 10-20 hPa pressure level. (b) Time series from May 1$^{st}$ to October 31$^{st}$ of the daily PSC density averaged over the 2006-2020 period (blue triangles for $\overline{F(P)}$, orange triangles for $\overline{F'(P)}$) with their standard deviation (vertical blue and orange lines). The blue and orange envelopes represent the maximum and the minimum daily densities. (c) 2D histogram of $\overline{F'(P)} - \overline{F(P)}$ differences vs $\overline{F(P)}$.**

The PSC densities simulated by our model $\overline{F'(P)}$, calculated as in Sect. 3.3 using one temperature threshold $T_{psc}$ and a set of regression parameters per month (Appendix E), are shown in red in Fig. 6. In October, there were no PSC observations at 10-20 hPa, so for that month we used parameters from September. Our model captures the PSC seasonal evolution well, but appears to miss short-lived peaks in PSC densities (e.g., 2009 and 2012). Calculating $T_{psc}$ for each altitude level over the 2006-2020 period appears to lead to poorer prediction performance in individual seasons, compared to determining these values for



a given year (as in Sect. 4). For instance, in June-July 2009 simulated PSC densities do not agree with observed ones as well as in Sect. 4.1. PSC densities averaged over the whole period, however, are well reproduced (blue and red lines in Fig. 6b) with a mean absolute error (MAE) of 0.005. Standard deviations of both $\overline{F'(P)}$ and $\overline{F(P)}$ follow a similar evolution and proportion, peaking at 0.06 in the middle of the season before decreasing. Our model has trouble reproducing the maximum

values of PSC densities illustrated by the mismatch of the blue and red triangles, as was already observed in Fig. 6a. The daily differences between $\overline{F'(P)}$ and $\overline{F(P)}$ can reach 7.5% (Fig. 6c) but only for high values of observed PSC densities ($> 0.2$). For low observed PSC densities ($< 0.05$), the error of our model is of the same order of magnitude, however the values being less than 10%, they remain negligible. 79% of simulated PSC densities are within ±0.02 of retrievals (Fig. 6c, red lines), and 96% within ±0.05 (orange lines). These results suggest that our model manages to reproduce the PSC density in the upper

stratosphere over a long-time period with appropriate accuracy.

We focus now on the start and end of the PSC season, according to PSC densities derived from observations and our model. In a given year, we mark the day when these densities first exceed 10% as the start of the PSC season, and the day when they last dip below this threshold as the end of the PSC season. PSC observations from 2006 were excluded from this part of the

analysis since PSC data are not available before June 13th, 2006. Figure 7a shows the start and end dates of PSC seasons according to those thresholds, considering PSC densities from observations (lines) and our model applied to temperatures (hatched regions). Both agree that at 10-20 hPa the PSC season is globally short, lasting approximately one month. On average over the 2007-2020 period, both agree that the season starts around the end of June (June 26th for our model and June 24th for observations) and ends on August 1st according to observations and on July 27th according to our model. In 2010, the PSC

density is too low to register on Fig. 7a. In 2012, the predicted PSC densities exceeded 10% only for two days, on July 2nd and July 3rd. Apart from these years, characterized by a warm vortex, the shortest season is in 2015, potentially due to the volcanic eruption of the Calbuco (Sect. 7), the PSC season lasts 4 days according to our model and 10 days according to observations. PSC season lengths from both PSC densities (Fig. 7b) are in very good agreement over 2007-2020, and appear very stable. A regression analysis finds a slight decrease in PSC season length but no significant trend compared to interannual variability.

Overall, the boundaries of the PSC season and its duration derived from our model and from observations agree well over the 2007-2020 period.



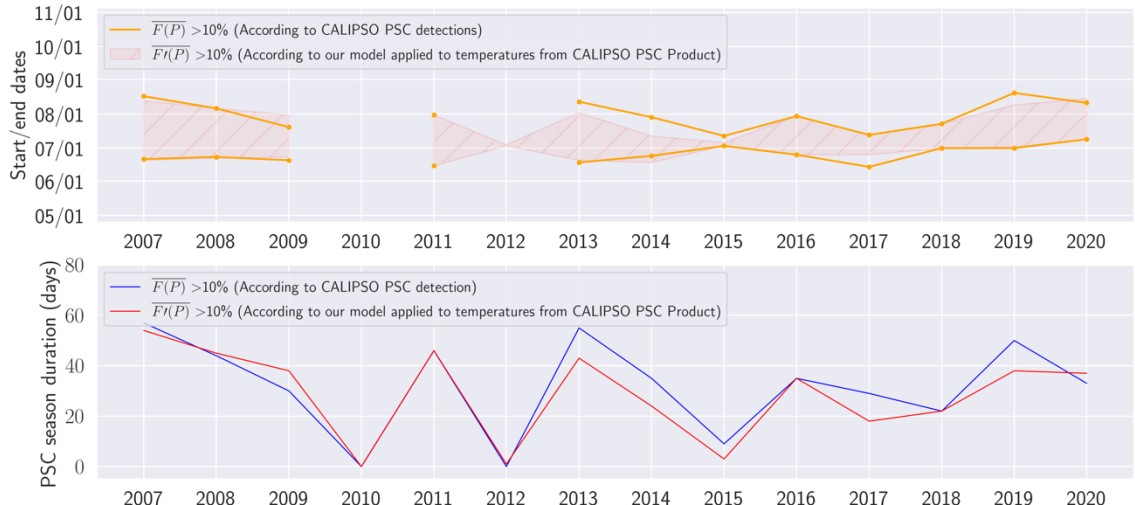

**Figure 7: (a) Start and end dates of the PSC season for the 10-20 hPa pressure level over 2007-2020. Full lines are based on PSC densities $\overline{F(P)}$ retrieved from CALIPSO PSC detections, hatched surfaces on PSC densities $\overline{F'(P)}$ derived from our model applied to MERRA2 temperatures contained in the PSC product. When PSC densities exceed 10% is shown in orange. (b) PSC season duration according to observations (blue) and to our model applied to MERRA2 temperatures contained in the PSC product (red) from 2007 to 2020.**

## 5.2 50-70 hPa pressure level

At the 50-70 hPa pressure level, observed PSC densities (Fig. 8a) are more important than at 10-20 hPa. Large $\overline{F(P)}$ are generally between 40% and 60%, and exceed 0.6 in 2006 due to a particularly large and cold vortex (WMO 2006, Pitts et al., 2018). The observed PSC densities are lowest in 2010, 2012 and 2015. On average over the 2006-2020 period (Fig. 7b), PSC are observed starting in late May and until early October. During the first 2 months of the season, the observed standard deviation remains low, below 0.08, suggesting that all PSC seasonal cycles start with the same pattern until July (Fig. 7b). In July, standard deviations increase and reach a maximum of 0.12 in late July and early August, before slowly decreasing until the end of the season. Largest densities are reached between July and August with values around 40%. Largest PSC densities are on average observed on July 16[th], two weeks before the coldest temperatures are reached (August 2[nd], not shown). This suggests that starting July 16[th], concentrations of $HNO_3$ and $H_2O$ have decreased enough to provide constraints on PSC formation, which is consistent with a colder $T_{psc}$ for the next months (Sect. 3.2).



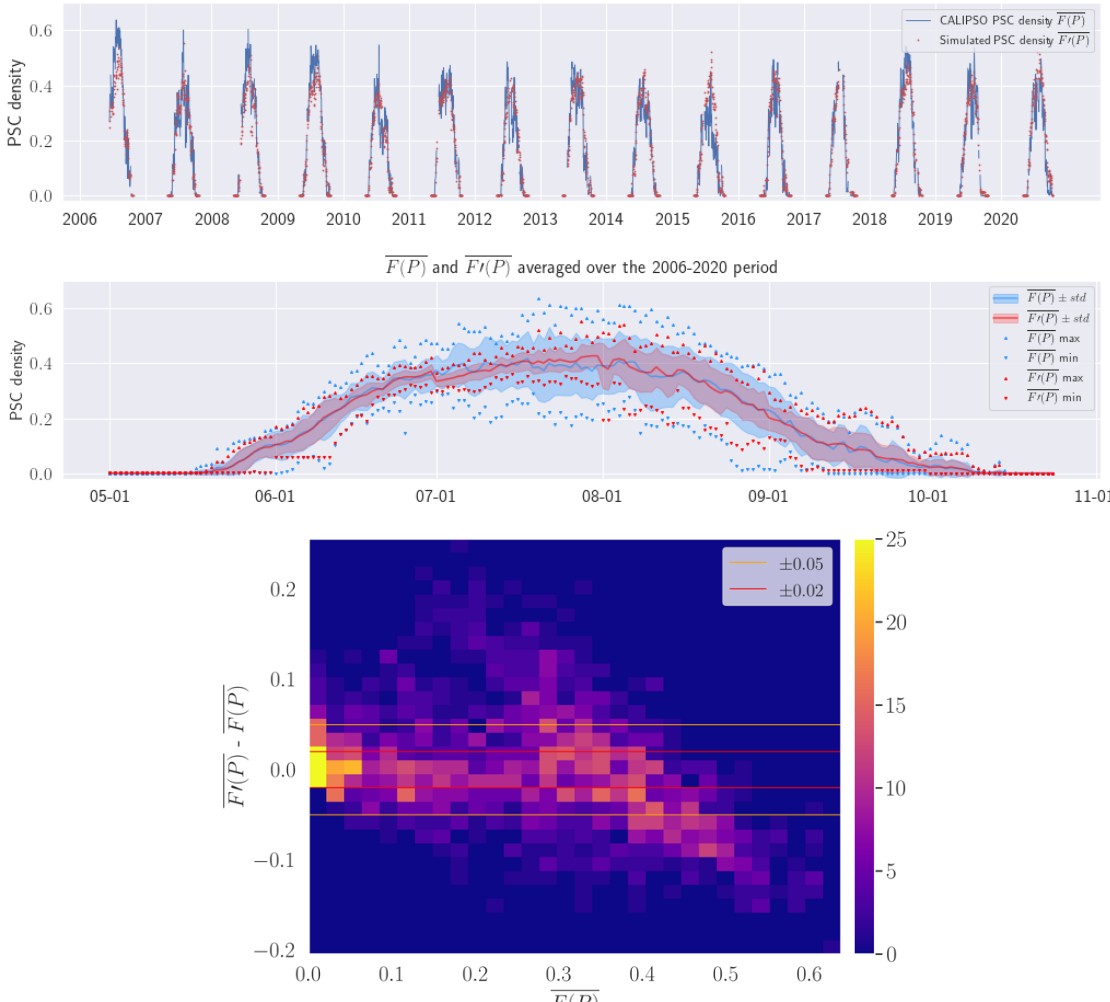

**Figure 8: Same as Figure 6 but for the 50-70 hPa pressure level.**

Our model represents well the PSC densities, except in July and August 2015 when $\overline{F'(P)}$ are larger than observations by up to 25% (Appendix F, Fig. F1). This year was characterized by the Calbuco eruption, which impacted stratospheric aerosol concentrations (Zhu et al., 2018). Such changes cannot be taken into account in our model which is temperature-based. We will discuss the consequences of eruptions more broadly in Sect. 7. Apart from 2015, at this pressure level, average PSC densities are reproduced extremely well (Fig. 7b), the MAE over the entire season between the average observed and simulated

PSC densities is 0.01. Our model performance is best in May, early-June and September-October: the discrepancy between average PSC densities observed and simulated are extremely low, and standard deviations from both our model and observations are the same (Fig. 7b). This means that PSC densities for those months are well correlated to the evolution of cold densities below the temperature thresholds chosen. Our model slightly underestimates PSC densities in early July and



slightly overestimates them at the end of July, as a consequence the MAE is slightly higher while remaining low (0.02). Overall,
48% of simulated PSC densities are within ±0.02 of retrievals, and 71% within ±0.05 (Fig. 7c, orange and red lines). For low
PSC densities (< 0.2), our model tends to overestimate the densities, with 19% of the differences between $\overline{F(P)}$ and $\overline{F'(P)}$
being > 0.02 versus 11% < -0.02. When the observed PSC densities increase (> 0.2), our model underestimates PSC densities
in a linear way with 46% of the differences between $\overline{F(P)}$ and $\overline{F'(P)}$ being < -0.02 against 30% > 0.02.

At the 50-70 hPa pressure level, according to observations, on average over the 2007-2020 period the PSC season starts on
May 31st, ends on September 16th, and lasts 108 days (orange line, Fig. 9a). The season lasts 67 days longer than for the 10-20
hPa pressure level, starting earlier and ending later. Our model, over the same period, predicts the same start and a slightly
earlier end date on September 12th (hatched, Fig. 9a). The notable differences in end dates between observations and our model
in 2011 and 2015 are due to peaks in observed PSC densities at the end of the season. Our model partially manages to reproduce
them by predicting PSC densities lower than 10%. A regression analysis reveals a shortening trend in the duration of the PSC
season according to both observations (blue, Fig. 9b) and our model (red, Fig. 9b). The trend is statistically significant only
for observations, according to which the season length decreased by 18 days between 2006 and 2020. The start date of the PSC
season remains relatively stable according to observations and our model. End dates show a significant decrease for both
observations and our model, moving from September 21st to early September.

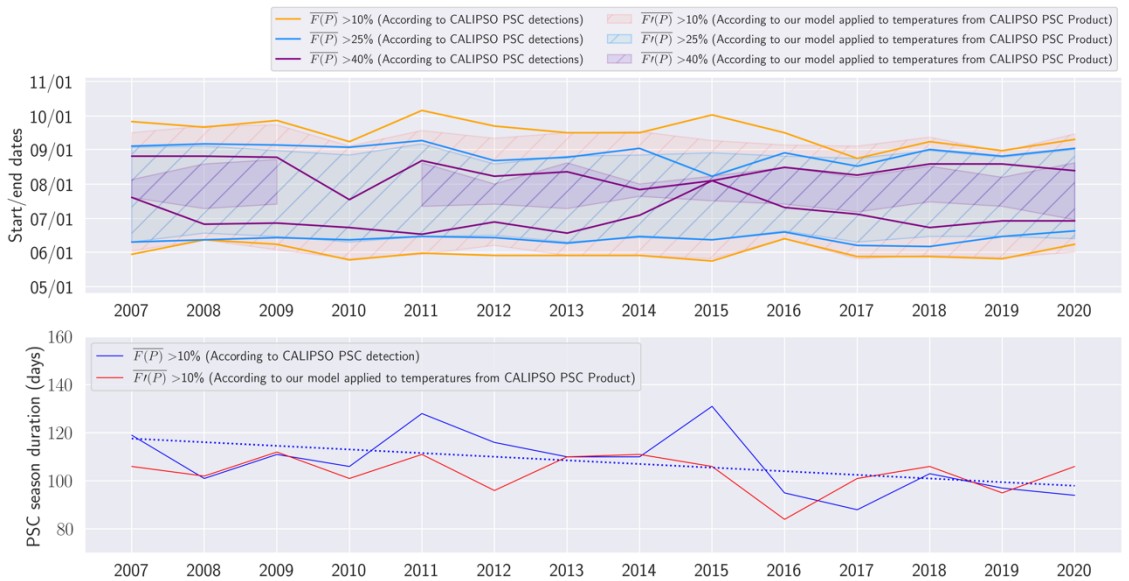


**Figure 9: Same as Figure 7 but for the 50-70 hPa pressure level.**

From now on we call P25% the period when PSC densities get larger than 25% (blue on Fig. 9a), similar to how PSC seasons
are defined (Sect. 5.1). According to observations, on average over 2007-2020, P25% starts on June 13th and ends on August



29th. According to our model (hatched blue in Fig. 9a), on average over 2007-2020, P25% starts one day later and finishes one day earlier than observations, thus our model and observations agree very well on the onset and end of P25%. However, in 2015, a disparity emerged between observations and our model regarding the end-of-season, with an earlier end date according to observations. Over the considered period, the duration of P25% is diminishing according to both observations and our model, displaying a statistically significant declining trend by 10 days between 2007 and 2020 (Appendix F, Fig. F3a). This

reduction is attributed to earlier end dates, shifting from early September to the end-August. Finally, we now call P40% any period of very large PSC densities (> 40%) (purple on Fig. 9a). On average over 2007-2020, observations put P40% between July 3rd and August 12th (43 days). While our model has trouble reproducing the early onset of high PSC densities (it predicts they start on average on July 14th), it accurately predicts their disappearance. Our model indicates PSC densities above 40%, except in 2010. There is no statistically significant trend of the P40% season according to either observations or model

(Appendix F, Fig. F3b).

## 5.3 100-150 hPa pressure level

At the 100-150 hPa pressure level, the PSC densities are less important than at 50-70 hPa but more important than at 10-20 hPa (Fig. 10a). In 2011, observed PSC densities are extremely low compared to other years, staying below 0.2 except during a few days in August. Those low densities are due to PSC NAT densities which are surprisingly low this year (Appendix G,

Fig. G1b), a result that to our knowledge has not been yet discussed in literature. Moreover, this year exhibits a clear delimitation between tropospheric and stratospheric clouds (Figure 13 in Pitts et al., 2018), while they overlap in other years, suggesting that the PSC densities observed at 100-150 hPa contain misclassified tropospheric cirrus. PSC density maximum of 35% is reached on August 5th, two weeks before the coldest day (August 16th). This is later than at 10-20 hPa (mid-July) and 50-70 hPa (July), as the stratosphere warms up from above. The observed PSC densities generally start to increase in mid-

June and finish in early October (Fig. 10b, blue). A prolonged PSC season is observed in 2006, 2011 and 2015, with PSC densities between 0.1-0.2 in early October (Appendix G, Fig. G1). Limited CALIPSO sampling in nighttime conditions beyond September prevents us from documenting with confidence the decrease in PSC densities after September, and thus the PSC season end date. PSCs are not uncommon over Antarctica later than September, as e.g. Lauster et al. (2022) documented PSC occurrence at 50 hPa as late as November from 1999 to 2021 at Neumayer (70°S) using ground-based observations.



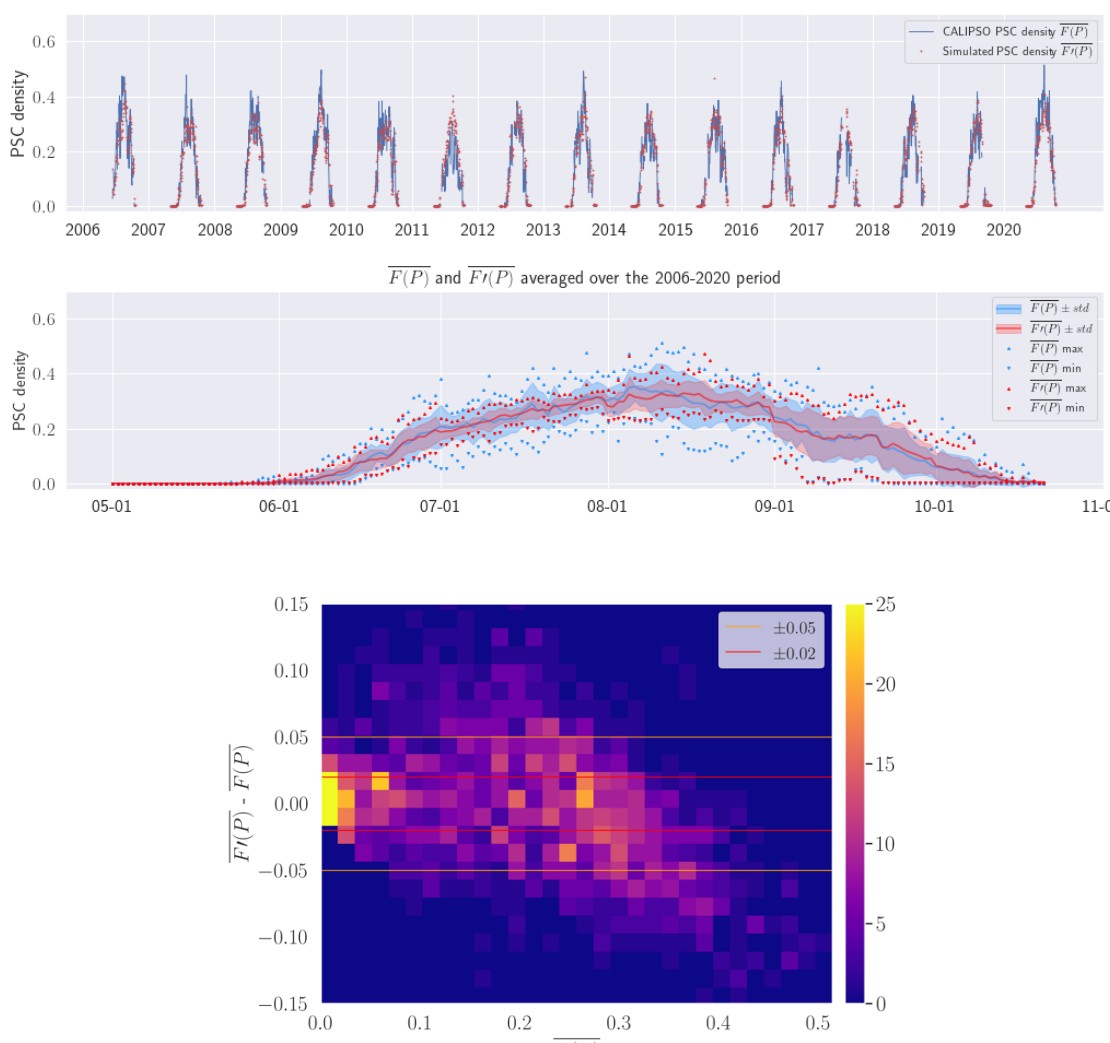

**Figure 10: Same as Figure 6 but for the 100-150 hPa pressure level.**

Overall, our model reproduces well the seasonal evolution of PSC densities, except in 2011, where it overestimates these densities throughout the season (Appendix G, Fig. G1). This discrepancy means that while the stratospheric temperatures are cold enough to generate more PSCs, their formation was somehow prevented. Average PSC densities are well predicted by our model (Fig. 10b), with a mean absolute error of 0.01 for the entire season. They are particularly well reproduced at the beginning (June) and end (September, early October) of seasons with negligible errors. However, in August, the MAE increases to 0.02 due to our model's underestimation of average PSC density in early August and overestimation for the rest of the month. At the onset and end of the season, our model captures well the inter-annual variability as illustrated by the match of the blue





and red envelopes, mostly noticeable in September and October. Moreover, our model captures maximas and minimas well in June and September. Model performances deteriorate in July and August, with missing maxima and minima, as well as standard deviations predicted by our model lower than those observed, like at 50-70 hPa. The interannual variations of the simulated PSC densities are minimal from May to September, with a standard deviation below 0.04, indicating stable MERRA2 stratospheric temperatures from one year to the next. However, from September the standard deviation increases twice as

much, meaning that stratospheric temperatures at the end of the season vary more (Appendix G2). We notice that these high standard deviations predicted by our model are also visible in observations, suggesting that MERRA2 represents well interannual variations in stratospheric temperatures in September and October. Overall, 53% of simulated PSC densities are within $\pm0.02$ of retrievals, and 79% within $\pm0.05$ (Fig. 10c, orange and red lines). For PSC densities between 0.1 and 0.2, our model overestimates the densities with 27% of the differences between $\overline{F(P)}$ and $\overline{F'(P)} > 0.05$. Like at 50-70 hPa, as PSC densities

increase ($> 0.3$), our model underestimates them in a linear way with 45% of the differences between $\overline{F(P)}$ and $\overline{F'(P)} < -0.05$ against 2% $> 0.05$.

At the 100-150 hPa pressure level, on average over 2007-2020, observations report the PSC season starts on June 18[th] and ends on September 27[th] (Fig. 11a). Our model is almost in perfect agreement with observations. At this pressure level the PSC season lasts for 3 months (102 days) on average, with our model and observations almost in perfect agreement (Fig. 11b). The

shortest season is observed in 2019 with a duration of 77 days. The season duration is very stable across the documented period, with no significant trend.

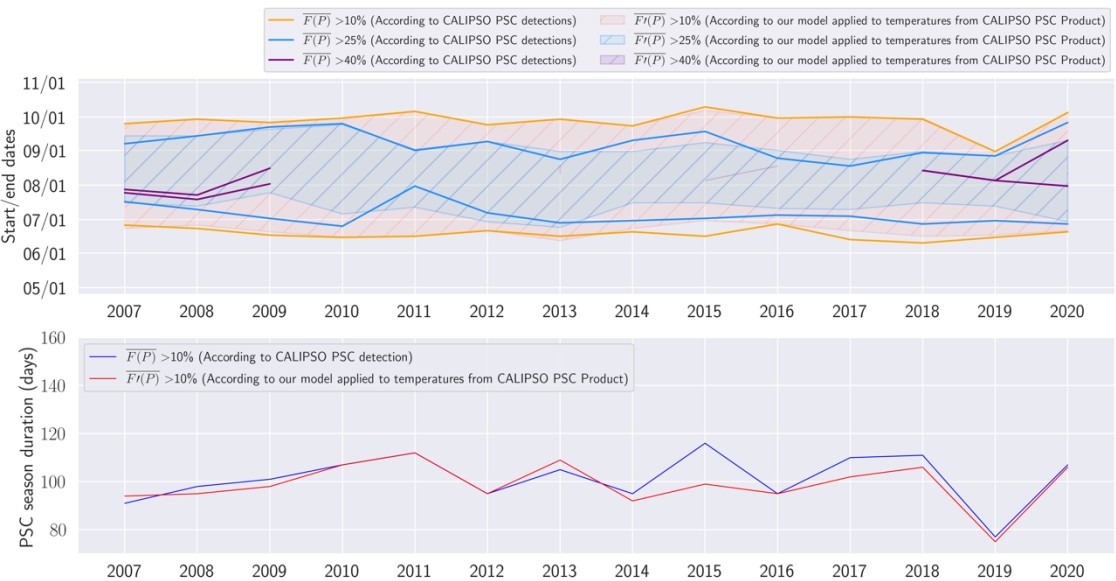

**Figure 11: Same as Figure 7 but for the 100-150 hPa pressure level.**




Observations indicate that, on average over the 2007-2020 period, PSC densities exceed 25% (P25%) between July 4th and
September 7th. Our model predicts P25% quite late (blue shading, July 10th on average) compared to observations (blue lines),
but is in perfect agreement with observations regarding the end date. According to observations, the P25% lasts 2 months on
average, with no significant trend (Appendix G, Fig. G3a). However, according to our model, the regression indicates a
statistically significant decrease of 13 days, due to an earlier end date, shifting from mid-September to end-August. PSC
densities rise rarely above 40%, from 2007 to 2009 and from 2018 to 2020 (purple lines, Fig. 11a). As indicated by the short
duration of P40%, these are sudden peaks in the observed PSC densities. Our model manages to reproduce the global pace of
PSC densities but misses the sudden peaks, explaining why our model only predicts PSC densities above 40% during one day
in 2009, 2011, 2015, 2016, during 9 days in 2013 and 3 days in 2020.

## 5.4 Model performance in predicting season start and end dates

Considering the start of the PSC season, our model agrees extremely well with observations, as differences never exceed 2-3
days (red, Fig. 12a). Our model agrees best with observations at 20-30 hPa, 50-70 hPa and 70-100 hPa, with differences being
on average less than a day with standard deviations between 2-4 days. The largest difference with observations is found at
100-150 hPa (2.4 days ± 4.5 days), but remains small compared to the length of seasons. Concerning the end of PSC season
(Fig. 12a, blue), our model puts it slightly too early but agrees well with observations at low altitudes (100-150 hPa). The
average differences vary from 0 to 4 days with standard deviations between 0.8 and 11 days at highest altitudes. Our model's
most accurate prediction is at 100-150 hPa, with a sub-day average and standard deviation of differences. Overall, the PSC
season dates produced by our model agree extremely well with observations, at the day scale below 50 hPa and week scale
above.



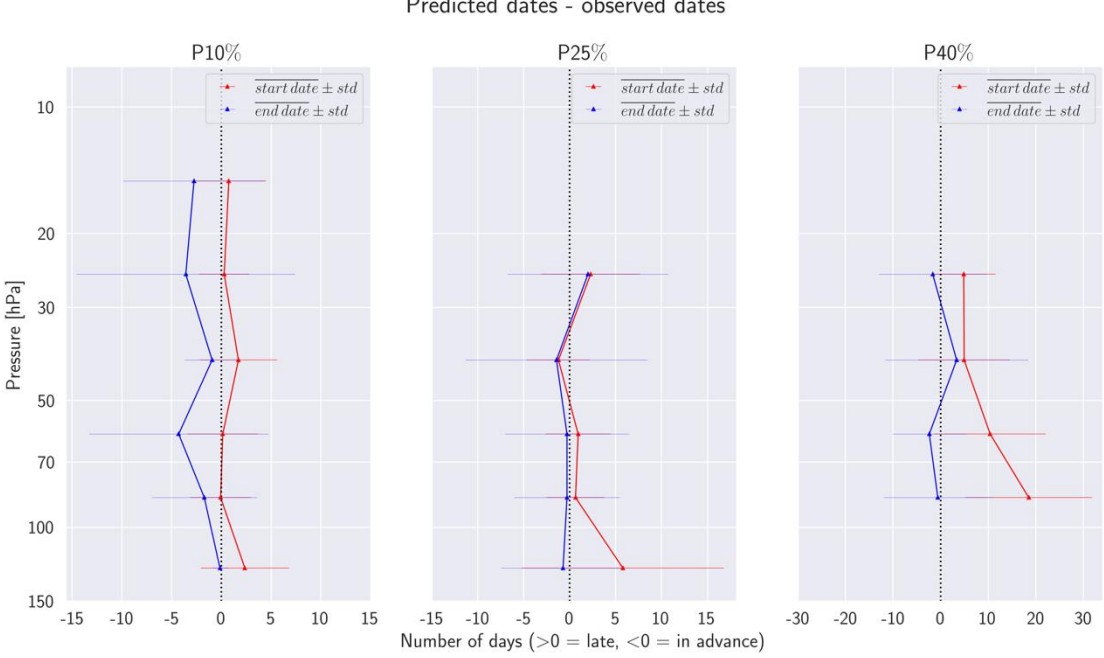

**Figure 12: Vertical profiles of the average differences between the start dates predicted by our model and observed (red curves) for**
**P10% (a), P25% (b) and P40% (c), over the 2007-2020 period. Similarly, the differences between the predicted and observed end**
**dates are represented by the blue curves. Only pressure levels with more than 10 years of comparison between observations and our**
**model are included in the plot. Our model is applied to the MERRA2 temperatures contained in the PSC product. The horizontal**
**bars represent the standard deviations of differences.**

Moving to P25%, our model predicts start dates late by up to 6 days compared to observations (red, Fig. 12b), except at 30-50
hPa where it predicts early start dates by around 1.2 days. Standard deviations for the differences in start dates range between
3 and 11 days. Similarly, P10% predictions are worst at 100-150 hPa (with an average difference of $5.8 \pm 11$ days). Predicted
end dates for P25% are generally close to observations (blue, Fig. 12b), with differences up to 2 days with standard deviations
between 5 and 10 days. The most accurate prediction for end dates is observed at 50-70 hPa and 70-100 hPa with an average
difference of $0.29 \pm 6$ days. The agreement of predicted end dates deteriorates at 20-30 hPa. While P25% prediction start dates
are slightly less accurate than P10%, they remain robust overall. However, the end dates predictions surpass P10% in accuracy.
Considering P40%, our model finds start dates consistently later than observations by 4 to 18.5 days (red, in Fig. 12c), with
standard deviations of differences between 7-13 days, the lowest one at 20-30 hPa (7 days). The best agreement is in the upper
stratosphere from 20 to 50 hPa, where the predicted start date is delayed by only 5 days compared to observations. P40% end
date predictions (blue in Fig. 12c) are generally ahead by 0.6 to 3 days, except at 30-50 hPa where it is late by 3 days, with
standard deviations of differences between 8 and 15 days. The least accurate end date prediction is at 30-50 hPa, with an



average difference of 3 ± 15 days. Our model struggles to predict robust start dates for P40%, because it does not capture well the peaks present in the observed PSC densities (Sect. 5.3). Its predictions for end dates agree better with observations.

In summary, the boundaries of PSC season and of P25% period predicted by our model agree extremely well with those derived from observations, with differences below 2-4 days which are small relative to the season duration. For P40%, the predicted start dates are late compared to observations, especially at lower altitudes where delays can exceed 10 days. In general, our model tends to slightly shorten the PSC season as well as P25% and P40% periods, due to later start dates and earlier end dates. Differences are however relatively small. As PSC seasons provided by our model appear consistent with the one derived

from observations over the period 2007-2020, we can now in the next section extend our analysis into the past using MERRA2 gridded temperatures (Sect. 2.2) over the 1980-2021 period.

## 6 Changes of the PSC season length over the 1980-2021 period

In this section, we investigate long-term changes in the length of the PSC season over the entire polar region (60°S-90°S) during the 1980-2021 period. To do so, we simulate daily PSC densities $\overline{F'(P)}$ (Sect. 3) using MERRA2 gridded temperatures

(Sect. 2.2), which cover the entire Antarctic region homogeneously every day. Since our model is built on the relationship between PSC detections from CALIPSO and MERRA2 temperatures at large spatial resolutions, it is directly applicable to the gridded temperatures from the same dataset. Compared to using temperatures from the CALIPSO PSC product, which sample different parts of the polar region every day (Sect. 4 and 5), this approach leads to a more complete representation of daily PSC densities that does not suffer from sampling degradation due to variations in instrument operation or incoming sunlight.


We assume that the PSC-temperature relationship we derived between 60°S-82°S is still valid south of 82°S. We focus on three pressure levels: 10-20 hPa (Sect. 6.1), 50-70 hPa (Sect. 6.2) and 100-150 hPa (Sect. 6.3).

### 6.1 10-20 hPa pressure level

At the 10-20 hPa pressure level, over 1980-2021, our model applied to MERRA2 gridded temperatures finds that the PSC

season (P10%) starts on average on June 17[th] and ends on August 12[th] (Fig. 13a), respectively 1 week earlier and 12 days later than observation-based analysis (Sect. 5.1). This longer period can be attributed to the gridded dataset including latitudes south of 82°S, absent from the CALIPSO dataset. The PSC seasons are particularly short in 2001 (one week, July 23[rd] -29[th]) and 2006 (one day). In those years, the average temperature at 10-2 0hPa stayed warmer than 188 K during the entire winter, with only short-lived, sporadic peaks in PSC densities above 10%. In 1996, 1998 and 2021, the end of season on August 3[st] is due

to a sudden drop in the PSC density, reflecting a rapid stratosphere warming. At 10-20 hPa simulated PSC densities almost never exceed 25% (blue hatch, Fig. 13a).




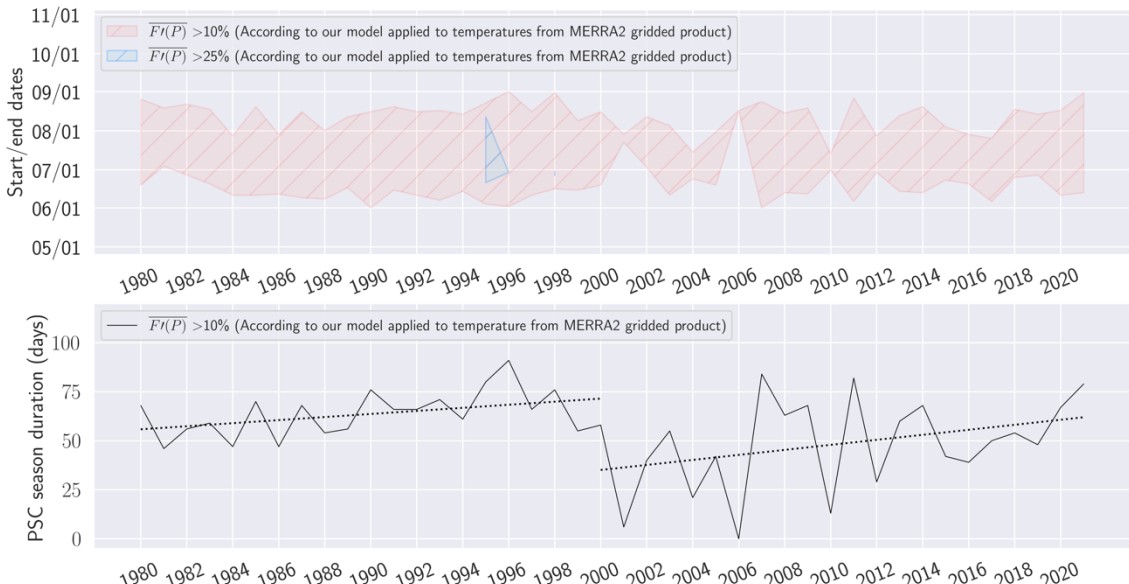

**Figure 13: (a) Start and end dates of the PSC season for the 10-20 hPa pressure level over 1980-2021, derived from our model applied**
**to the MERRA2 gridded dataset. When PSC densities exceed 10% is shown in orange, 25% in blue. (b) PSC season duration at 10-**
**20 hPa according to our model applied on MERRA2 gridded temperatures from 1980 to 2021. Dashed lines represent regressions.**

Two distinct periods are visible in the duration of the simulated PSC season (Fig. 13b). In the first period (1980-2000, dashed
black line), the PSC season got longer by 15 days. This increase in duration is due to an earlier start and a later end of the PSC
season between 1980-2000 (Fig. 13a). Our model being based on temperature, this increasing trend between 1980 and 2000
necessarily reflects that over that period the stratosphere stayed colder than the formation threshold for increasingly long
periods. The length of the PSC season drops dramatically in 2001 and remains low for several years. In 2007 it starts increasing
again, expanding by 27 days between 2000 and 2021. In spite of the two statistically significant positive trends over the 1980-
2000 and 2000-2020 periods, the important drop in PSC season length over the 2000-2006 period explains why no significant
trend is found over the entire 1980-2020 period.

## 6.2 50-70 hPa pressure level

At the 50-70 hPa pressure level, over 1980-2021, our model find that the PSC season starts on average on May 30[th] and ends
on September 20[th] (orange hatched, Fig. 14a). In 1988, 2002 and 2019, the PSC season ended early, illustrating sudden
stratospheric warming (SSW) for those years (Sect. 7). Using gridded temperatures unconstrained by sampling limits enables
a more reliable prediction of when the PSC season ends. The start date of the PSC season appears stable over the 1980-2000
period. However, seasons appear to end later in more recent years. Regression analysis finds a significant positive trend in the



length of the PSC season over the 1980-2021 period (Fig. 14b), expanding by 20 days. A particularly important trend is found in the 1980-2000 period, linked to a significant increase right after 1992, following the Mount Pinatubo eruptions (Sect. 7). From 2001 to 2021, the season length appears stable but its interannual variability is large.

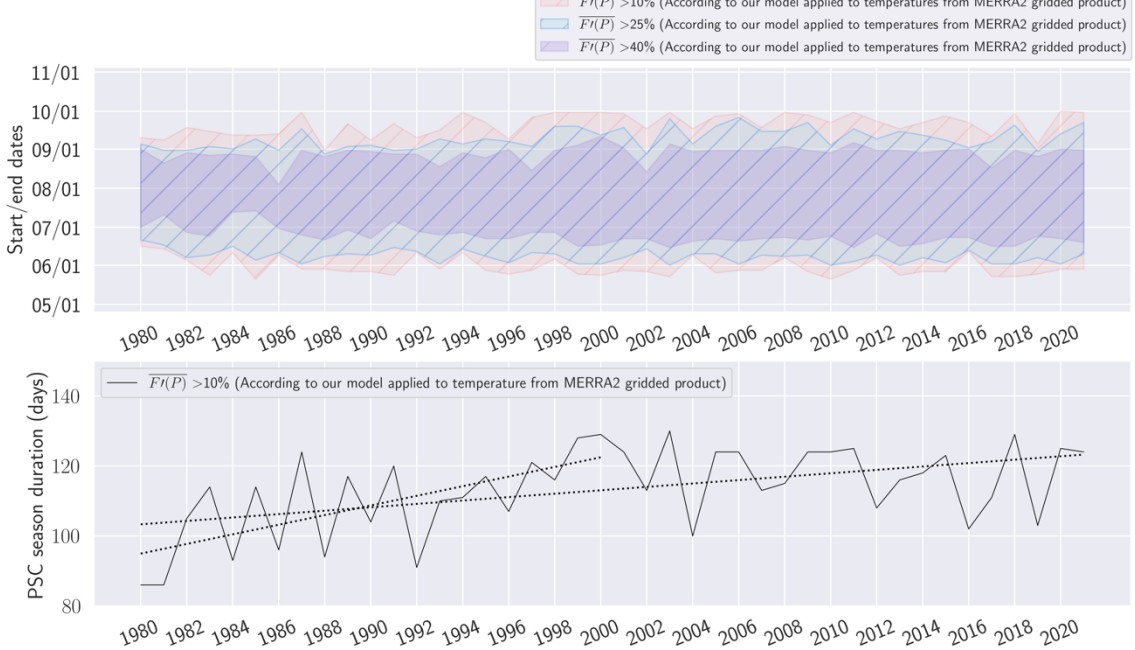

580

**Figure 14: Same as Figure 13 for the 50-70 hPa pressure level.**

Our model predicts that over 1980-2021 the P25% period starts on average on June 7th and ends on August 28th. Regression analysis shows a statistically significant increase over the 1980-2021 period and as well as 1980-2000 period, both expanding

585 by 20 days (Appendix F, Fig. F3a). Our model predicts the P40% period starts on average on June 24th, and ends on August 29th, notably longer than observation-based regressions (Sect. 5.2). About the duration of P40%, a regression shows once again a statistically significant increase over the 1980-2021 period and 1980-2000 period, both expanding by 20 days (Appendix F, Fig. F3b). After 2000, the P40% period is stable, and lasts around 70 days.

## 6.3 100-150 hPa pressure level

590 At the 100-150 hPa pressure level, over 1980-2021, our model find the PSC season starts on average on June 11th, and ends on October 9th (orange hatched, Fig. 15a). End dates are most of the time predicted in October, with the latest (October 30th) being in 1995 and 2021. Over 1980-2021 the length of the PSC season is stable (except for a marked minimum in 2002) and lasts on average 120 days, with no statistically significant trend (Fig. 15b). However, over the 1980-2000 period (dashed line



in Fig. 15b), a statistically significant increase expands the season length by 30 days due to later end dates, going from early October to end-October. The year 2002 and 2019 exhibits early end dates due to SSW (Sect. 7).

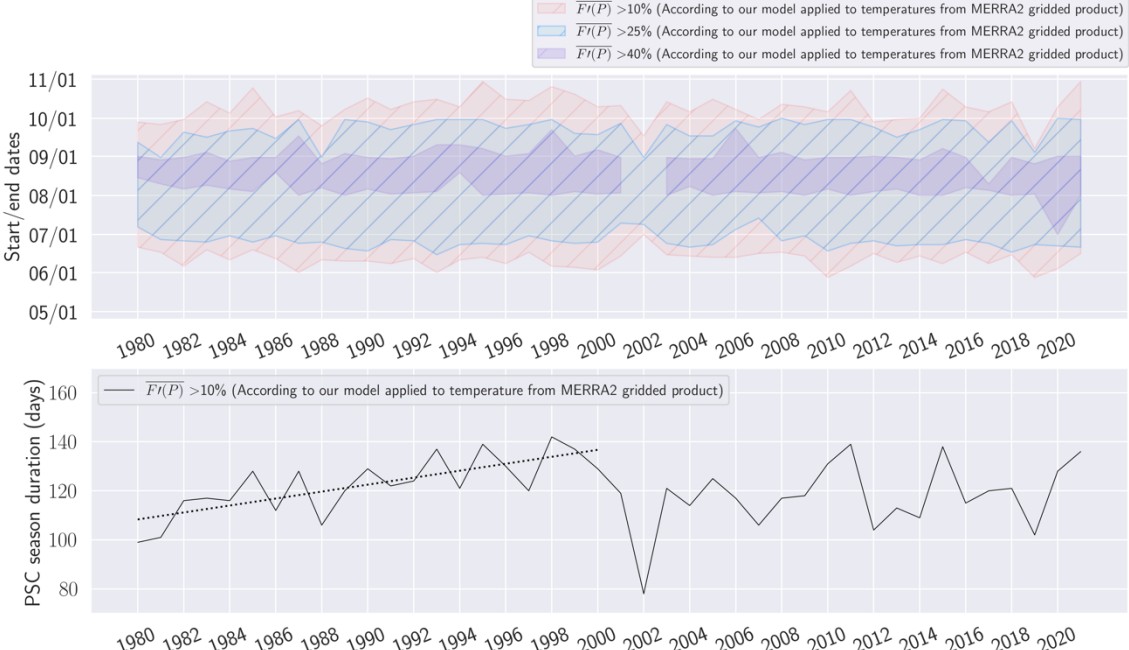

**Figure 15: Same as Figure 13 for the 100-150 hPa pressure level.**

Our model predicts the P25% period starts on average around June 26<sup>th</sup> and ends around September 22<sup>nd</sup>. Over the 1980-2021 period the P25% period lasts on average 88 days (Appendix G, Fig. G3a) with no statistically significant trend. However, over the 1980-2000 period, a statistically significant increase expands the P25% period by 20 days. Our model finds the P40% period starts on average on August 4<sup>th</sup> and ends on September 2<sup>nd</sup>. The P40% onset and end-of season are stable over the 1980-2021 period. However, the P40% period ends quite late in 1987, 1998 and 2006, and starts early in 2020. Regression analysis finds a statistically significant increasing trend between 1980 and 2000, expanding by 1 week (Appendix G, Fig. G3b), but no statistically significant trend over the 1980-2021 period.

## 6.4 Changes in PSC season duration over all pressure levels

Here, we outline the evolution of the PSC season length (P10%) according to time series trends for each pressure level over two periods: 1980-2000 and 1980-2021. Over the 1980-2000 period, we find the PSC season gets significantly longer (red line, Fig. 16a), in particular below 30 hPa. This increase exceeds the standard deviation of the series (dark green, Fig. 16a) at all pressure levels and exceeds twice the standard deviation (light green) between 30-150 hPa. The increase is largest at 70-





100 hPa, with an extension of more than a month (45 days). The lengthening of the PSC season is more prominent in the middle and lower stratosphere, suggesting a greater cooling trend in stratospheric temperature at these altitudes. The standard deviations between 1980-2000 show minimal variation across pressure levels, with a mean value at 9.5 days, suggesting stable

MERRA2 temperatures throughout the stratosphere during this period.

Over 1980-2021, changes in PSC season lengths are more modest (20 days at most, red line, Fig. 16b), and only exceed the standard deviation of the series between 30 and 100 hPa (dark blue, Fig. 16b). The change in PSC season duration is largest between 50 and 70 hPa, with an increase of 18 days, almost above twice the standard deviation (dark blue, Fig. 16b). At 10-

20 hPa and 20-30 hPa, the PSC season tends to get shorter, where it was found to get longer over the 1980-2000 period. This result illustrates the rapid warming after 2000 at these high altitudes. The lengthening of the season is less significant than over the period 1980-2000 due to a greater variability in the series, suggesting that after 2000, the polar stratosphere experiences greater temperature fluctuations towards warmer temperatures.

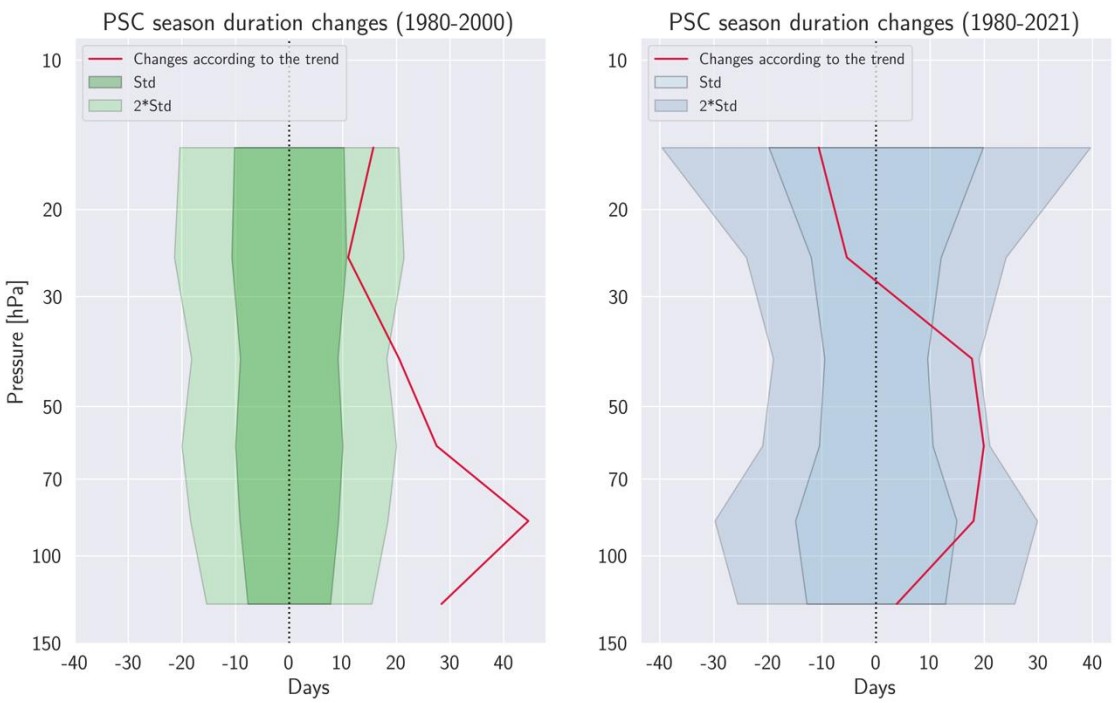


**Figure 16: PSC season duration changes between 1980-2000 (a) and 1980-2021 (b) for each pressure level. The lines represent the changes in the number of days in the PSC season duration according to regressions. The envelopes represent the standard deviations associated with the time series of the PSC season duration.**





## 7 Influence of synoptic events affecting the stratosphere

Large-scale atmospheric waves moving upwards into the stratosphere can disrupt the polar vortex. In extreme cases, this disruption triggers a reversal of wind directions and a rapid warming, known as a sudden stratospheric warming (e.g. SSW, Butler et al., 2015). In the Southern Hemisphere SSW events are rare, approximately one in 22 years (Jucker et al., 2021). The first SSW observed over the Antarctic was in 1988 with two distinct warming events (Roy et al., 2022). The first one lasted a few days from August 25[th] to September 3[rd], and was followed by a rapid cooling. The second one occurred from September

24[th] to October 3[rd] and was more intense than the first, witnessing a temperature rise from 205 K to 245-250 K within a week. In our result, we find for the year 1988 the earliest end date of P10% at 50-70 hPa (Fig. 14a) as well as early end date of P10% and P25% at 100-150 hPa (Fig. 15a). The second SSW in 2002 was characterized by multiple waves events that led to the warming of the polar lower stratosphere (Newman and Nash, 2005) and split the polar vortex. The warming started in June and became extremely intense by early July. In our results, we find at 50-70 hPa, low PSC densities (< 0.2) in June compared

to other years (not shown). Additionally, in the lower stratosphere at 100-150 hPa, we observe an early end date for P10% and P25% as well as no PSC densities above 40% (Fig. 15b). The P10% season ends on September 17[th] and the P25% season ends on August 31[th]. 2002 exhibits the shortest season at 100-150 hPa, lasting only 80 days against a season average at 120 days. Moreover, we observe a delayed onset of P10% and P25% compared to other years, starting at the beginning of July. A third SSW was recorded above Antarctica in late August-early September 2019, characterized by an exceptionally strong planetary

wave and a major warming (Yamazaki et al., 2020, Lim et al., 2021). MERRA2 data reported an exceptional rise in stratospheric temperature of 50.8 K per week at the 10 hPa level from September 5[th] to 11[th]. By comparison, the maximum warming during September 2002 was 38.5 K/week. In our results, at 10-20 hPa, this warming did not impact our PSC season since it ends before the beginning of the warming. However, at 50-70 hPa, we notice a significant decrease in simulated PSC densities since late August, consistent with the warming trend. P10% and P25% end early, on September 5[th] and August 30[th],

respectively (Fig. 14a). Moreover, at 100-150 hPa, our model finds the earliest end date for P10% on September 7[th] (Fig. 15b). We also find an early end date for P25% on September 4[th]. Based on these findings, we conclude that our model captures SSW events well as long as their influence appears in the reanalyses.

        In addition to SSW, the polar stratosphere can be affected by volcanic events. The eruption of Mount Calbuco, in April 2015

(Stone et al., 2017), injected into the stratosphere water and sulfate aerosols, which slowly descended during transport from mid-latitudes towards the poles. These injections began penetrating the Antarctic vortex in May, causing strong early denitrification and thus limiting PSC formation for the rest of the season (Zhu et al., 2018). CALIOP observations during May reveal high PSC densities at 50-70 hPa, marking the earliest onset of the PSC season on May 24[th] (Fig. 9a). These early PSC densities are made of NAT (Appendix F, Fig. F1) and are followed by a decline in early-June, providing confirmation of early

denitrification. Moreover, at 50-70 hPa and 100-150 hPa, we found low STS and NAT densities (< 0.2) during the middle of the season, unlike other years where densities are twice as high. The absence of high PSC STS and NAT densities at this period





suggests that there is no longer enough $HNO_3$ and $H_2O$ available in the stratosphere due to the early denitrification. In 2015, our temperature-based model overestimates the observed PSC densities at 50-70 hPa in July and August. The aerosols injected by volcanic eruptions capture part of the solar radiation, warming the atmosphere near their height of injection but cooling the

lower altitudes. MERRA2 temperatures at 50-70 hPa and 100-150 hPa (Appendix F2, G2), show a colder stratosphere than the 1980-2021 mean temperature from mid-July toward October probably due to the cooling effect of the sulfate aerosols. Another notable result, is the prolonged PSC season in observations at 100-150 hPa, showing densities around 0.2 in early October. Our model during this period underestimates the PSC densities, suggesting that factors beyond temperature influence the PSC formation, such as the aerosol supply resulting from the eruption. Considering PSC detections from CALIOP observations

over 2006-2020, SSWs have a much stronger influence on the length of the PSC season than volcanic eruptions.

While our model can take into account changes in stratospheric temperature due to the radiative effects of volcanic eruptions, it cannot take into account their impacts on stratospheric polar chemistry. Such changes will depend on the volcano location and distance to the poles, on the eruption intensity and date, and on which chemical compounds are released by the eruption

to reach the polar stratosphere. For example, the Hunga Tonga-Hunga Ha'apai eruption in 2022 injected huge amounts of water vapor into the stratosphere, but they did not penetrate the polar vortex, as revealed by satellite trace-gas measurements (Manney et al., 2023). Within the vortex, water vapor, like ozone and chemical species related to ozone thinning, maintained levels close to historical averages through spring 2022. Complications of the calibration of the CALIPSO signal in the stratosphere affected by the Hunga Tonga explains why the PSC product is at this time not available for the 2022 Antarctic winter, and why we

can't yet conclude about the effect of this eruption on the PSC formation and season. Outside the CALIPSO period and inside the 1980-2021 period, other volcanic eruptions include El Chichon (17°N, 93°W) in 1982, and Mount Pinatubo (15°N, 120°E) as well as Mount Cerro Hudson (46°S, 73°W), both in 1991. The Pinatubo eruption between June 12 and June 16, 1991, was the most significant in the past century, releasing sulfur dioxide ($SO_2$) into the stratosphere, which was transformed into sulfate aerosols ($H_2SO_4$/ $H_2O$) (McCormick et al., 1995) and affected mostly the Southern Hemisphere. These eruptions had an impact

on the global stratospheric temperatures. MERRA2 shows an episodic temperature increase between 10 and 100 hPa associated with these two volcanic eruptions (Gelaro et al., 2017). The cooling of the global mean stratosphere was more pronounced over the 1979-1998 period due to intense ozone depletion and the increasing GHG emissions compared to the 1998-2016 period where ozone depleting substances decreased (Maycock et al., 2018). Those results are consistent with the evolution of the PSC season duration simulated by our model in Sect. 6. Indeed, our model finds that between 1980 and 2000 the PSC

season gets longer at all pressure levels, consistent with a global cooling of the stratosphere. After 2000, the PSC season duration is stable, consistent with relatively warm years. To our knowledge, there is no evidence in the literature that confirms a PSC season extension over the 1980-2000 period, or that suggests the Pinatubo eruption led to early denitrification like with the Calbuco eruption in 2015. Fromm et al., (2003) made a unified stratospheric aerosol and cloud database from 1979 to 2000 by merging retrievals from several satellites. They found that the probability of detecting a PSC at 20 km in the Antarctic

vortex in July was particularly low in years characterized by volcanic eruptions, such as after the eruptions of El Chichon,




Nevado del Ruiz, and Mount Pinatubo. Moreover, those years were characterized by a strong aerosol extinction inside the Antarctic vortex. These results suggest that in case of major stratospheric eruptions, enhanced aerosol loading inside the polar vortex could trigger early denitrification and by consequence less PSC formation in the middle of the season, like for the Calbuco eruption in 2015. However, this does not imply a shortening of the PSC season, but only lower PSC densities during

the middle of the season. In this case, P25% and P40% are the results that will potentially be the most affected by volcanic eruptions.

## 8 Discussion and conclusions

In this article, we set out to evaluate if the seasonal evolution of PSCs had gone through significant changes over the past decades (1980-2021). To reach this objective, we defined an indicator to track the extent of daily PSC cover over the polar

region called PSC density. The PSC density represents the fraction of the volume covered by PSCs between two pressure levels. By relating the PSC density to temperature data from MERRA2, we developed a statistical model which predicts by pressure level the daily average PSC density over the polar region. We evaluated the robustness of our model by conducting comparisons between the observed PSC density data from CALIPSO and those simulated by our model.

The PSC-temperature relationship proposed by our model tracks well the daily evolution of PSCs throughout each season and at each pressure level between 10 and 150 hPa. Our model manages to reproduce the observed evolution of PSC in case of well-understood stratospheric temperature changes driven by SSW (e.g. 2019, Sect. 7). However, if radiative forcing by aerosols from eruptions is not accounted for in stratospheric temperatures, the PSC-Temperature relationship can be disrupted and our model will deviate from reality. Moreover, variations in aerosols and chemical species abundance following volcanic

eruptions, can affect our model's performance (e.g. the Calbuco eruption in 2015) in hard-to-predict ways due to the complex nature of their impact on PSC formation. Nevertheless, temperature generally appears to be a robust proxy for PSC formation at daily scales in most seasons, which enables a realistic capture of long-term changes. By applying our model to MERRA2 temperatures in the PSC product, we estimated when the PSC seasons start and end (the P10% period) over the CALIPSO record, and found a very good agreement with PSC observations, with an accuracy better than 5 days at all pressure levels.


Applying our model to gridded MERRA2 temperatures enabled us to quantify the PSC density 1) considering the whole Antarctica domain 2) at daily scales and 3) over the 1980-2021 period. This leads to an integrated view of PSC occurrence that is free from the sampling issues that spaceborne measurements can incur. Analyzing this dataset shows the length of the predicted PSC season is systematically longer than when using temperature from the CALIPSO dataset. We find that the PSC

season gets significantly longer during the 1980-2000 period, across all pressure levels with the largest increase (+ 45 days) at 70-100 hPa. At high altitudes, the prolonged PSC season is attributed to an earlier start date and a later end date, while at other



pressure levels, the lengthening of the season is due to later end dates only. Over the 1980-2021 period, between 30-100 hPa, the PSC season gets longer by 18 days on average. This lengthening is linked to a period of general stratospheric cooling. After 2000, the PSC season length dropped sharply at 10-20 hPa, 20-30 hPa and 100-150 hPa, a sign of stratospheric warming, then

got progressively longer again until 2020.

Our study associates observed PSC cover with MERRA2 temperatures. Applying our model to temperatures from other sources (e.g. ERA5 or climate models) would require re-training the model to account for baseline shifts in stratospheric temperatures for a particular dataset. Moreover, CALIPSO did not sample Antarctic regions south of 82°S, and variations in the PSC-

temperature relationship in that region can not be taken into account. There is however, to our knowledge, no evidence for such variations. Finally, events of regional-scale gravity wave cooling, which might be underrepresented in reanalyses, can trigger PSC formation (Noel and Pitts., 2012). These PSCs however constitute a relatively small fraction of the overall PSC cover and their impact on the model's performance is considered negligible.

Future work involves extending the scope of the present study to PSCs from the Arctic region, where stratospheric temperatures are less stable and PSC season durations more variable. First results suggest that our model can reproduce the PSC densities observed over the North pole over the 2006-2020 period as well as over Antarctica. Adding PSC speciation to our model would enable to study the evolution of a particular PSC type over long periods. Moreover, the gridded MERRA2 temperatures are available four times a day, so we could document the PSC density evolution at sub-daily scales. Finally, we intend to apply

our model to stratospheric temperatures predicted by general circulation models over the next century, to better understand the evolution of PSC cover in the context of climate change and how it could impact the reformation of the hole in the ozone layer.

**Data availability**

The CALIPSO/CALIOP L2 PSC Product dataset available at https://www-calipso.larc.nasa.gov/resources/calipso_users_guide/data_summaries/psc/index.php

The MERRA-2 gridded dataset reanalysis is available at
https://disc.gsfc.nasa.gov/datasets/M2I6NPANA_5.12.4/summary?keywords=

**Competing interests**

The contact author has declared that none of the authors has any competing interests.



## Acknowledgments

The authors thank Michael Pitts and the CALIPSO team of the Langley research center for suggesting paths of exploration for this article. The authors also thank the CNES and CNRS for supporting this work through the funding of the EECLAT project. This study has been supported through the grant EUR TESS N°ANR-18-EURE-0018 in the framework of the Programme des Investissements d'Avenir.

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

## Appendix A : CALIPSO sampling

The total daily number of CALIPSO profiles over a PSC season (May to October) from 2006 to 2020 at 10-20 hPa exhibits a yearly pattern, which illustrates the deterioration of CALIPSO PSC sampling as the season ends, due to the return of sunlight during Austral spring. Some days in the middle of the season exhibit a low number of CALIPSO profiles, which can correspond to CALIOP satellite maneuvers. In such cases, the payload is turned off, leading to irregular sampling and days excluded from the time series.

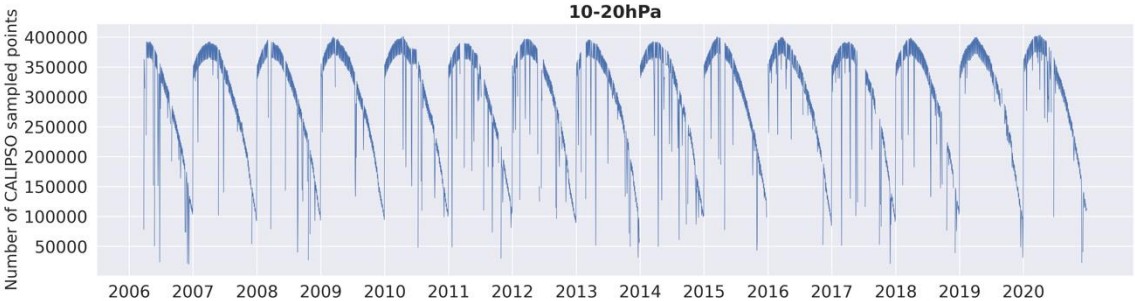

**Figure A1: Total daily number of CALIPSO points from 2006 to 2020 at 10-20 hPa.**

We sum up below, the percentage of available CALIPSO files from 2007 to 2020 for each month of the PSC season (May to October).

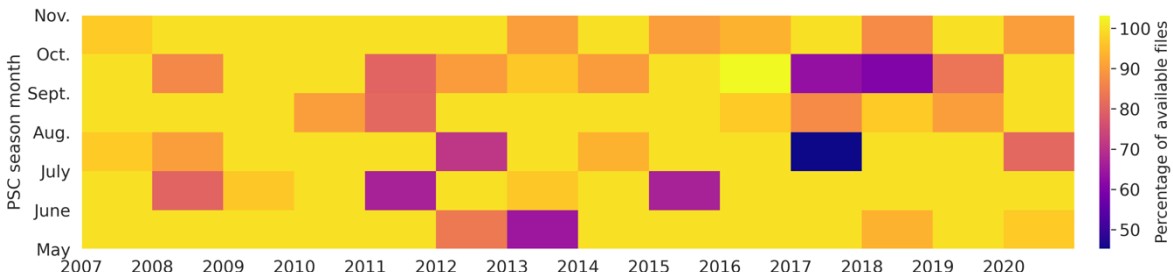

**Figure A2: Percentage of available daily files for each month of the PSC season from 2007 to 2020.**



## Appendix B : 2009 season

| Pressure [hPa] | $T_{psc}$ [K] | | | | | |
|---|---|---|---|---|---|---|
| | May | June | July | Aug. | Sept. | Oct. |
| 10-20 | 188 | 187 | 187 | 188 | | |
| 50-70 | 194 | 193 | 192 | 188 | 191 | |
| 100-150 | | 199 | 194 | 195 | 195 | 193 |

Table 1: Monthly temperatures thresholds $T_{psc}$ for the 2009 season. Empty boxes means that no $T_{psc}$ could be retrieved because there are no PSC observations during those months.

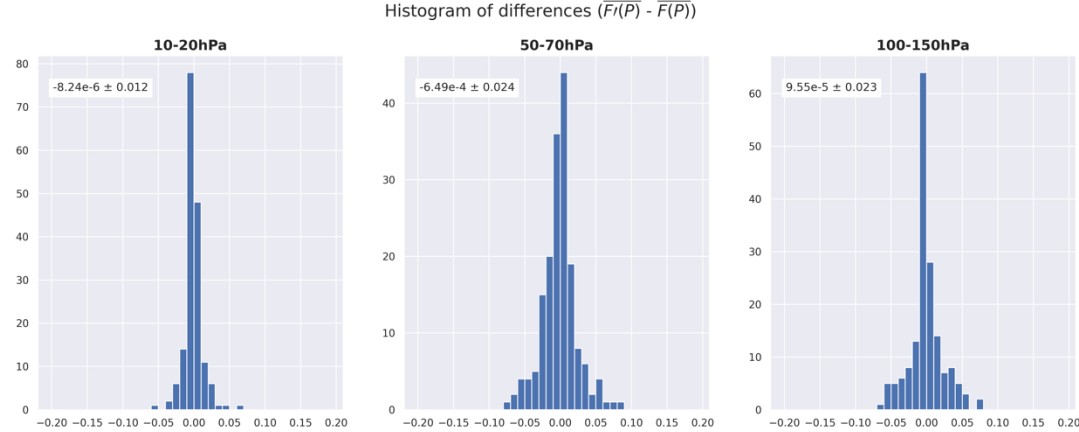

Figure B1: Histogram of differences between $\overline{F(P)}$ and $\overline{F'(P)}$ for 2009 at 10-20 hPa (left panel), at 50-70 hPa (middle panel) and 100-150 hPa (right panel).

## Appendix C : 2010 season

| Pressure [hPa] | $T_{psc}$ [K] | | | | | |
|---|---|---|---|---|---|---|
| | May | June | July | Aug. | Sept. | Oct. |





| 10-20 | 188 | 187 | 187 | 188 |     |  |
|-------|-----|-----|-----|-----|-----|--|
| 20-30 | 187 | 187 | 193 | 187 | 190 |  |
| 50-70 | 190 | 192 | 192 | 188 | 191 |  |

**Table 2: Monthly temperatures thresholds $T_{psc}$ for the 2010 season. Empty boxes means that no $T_{psc}$ could be retrieved because**
**there are no PSC observations during those months.**

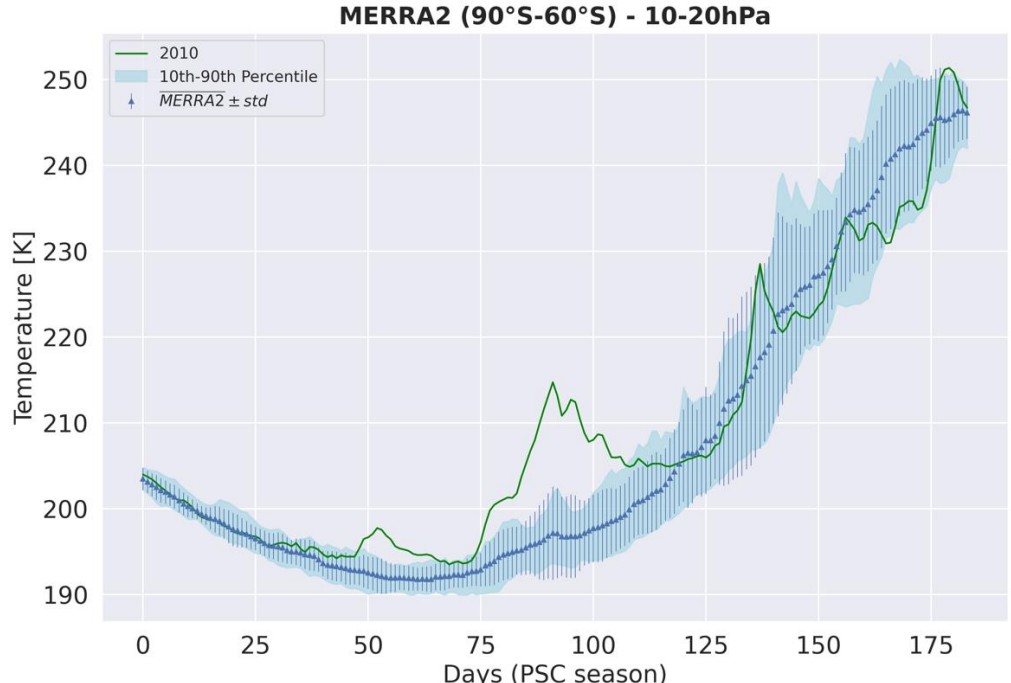

**Figure C1: Daily MERRA2 gridded stratospheric temperature average between 60°S-90°S and over each PSC season (2006-2020)**
**at 10-20 hPa. The time series goes from May 1st (day 0) to October 31st (day 184). The mean of MERRA2 (blue triangles) and**
**its standard deviation (blue vertical lines). The green line represents the year 2010.**





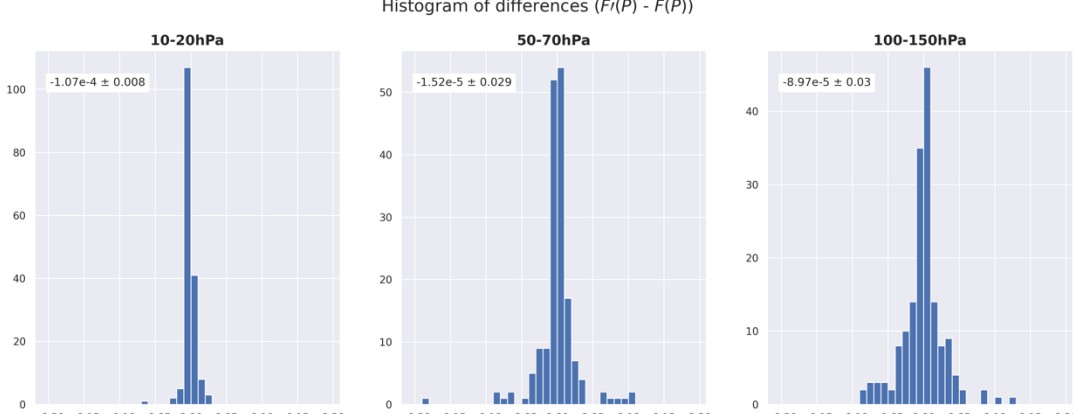

**Figure C2: Histogram of differences between $\overline{F(P)}$ and $\overline{F'(P)}$ for 2010 at 10-20 hPa (left panel), at 20-30 hPa (middle panel) and**

**50-70 hPa (right panel).**





## Appendix D : Monthly scatterplots over the 2006-2020 period at 50-70 hPa

**Figure D1: Monthly scatterplots for the 50-70 hPa pressure level over 2006-2020. Red lines are the trend lines with their equations in the left white panel as well as the correlation coefficients $R^2$.**





## Appendix E : Monthly temperatures thresholds and statistical parameters (2006-2020)

| Pressure [hPa] | $T_{psc}$ [K] | | | | | | $MAE$ | $RMSE$ | $R^2$ |
| | May | June | July | Aug. | Sept. | Oct. | | | |
|---|---|---|---|---|---|---|---|---|---|
| **10-20** | 188 | 188 | 188 | 187 | 190 | 190 | 0.01 | 0.02 | 0.86 |
| **20-30** | 189 | 188 | 190 | 188 | 188 | 188 | 0.03 | 0.05 | 0.91 |
| **30-50** | 192 | 189 | 193 | 188 | 187 | 187 | 0.03 | 0.05 | 0.92 |
| **50-70** | 193 | 191 | 194 | 192 | 188 | 188 | 0.04 | 0.05 | 0.90 |
| **70-100** | 196 | 195 | 195 | 195 | 190 | 192 | 0.03 | 0.05 | 0.90 |
| **100-150** | 198 | 198 | 196 | 196 | 193 | 192 | 0.03 | 0.04 | 0.90 |

**Table 3: Monthly temperature thresholds $T_{psc}$ by pressure level $P$ and statistical parameters derived from comparisons between**
**PSC densities observed by CALIPSO and those simulated by our model over the 2006-2020 period.**

MAE (mean absolute error) is a measure of the prediction difference between two time series.

$$MAE = \frac{\sum_{days}|\overline{F(P)} - \overline{F'(P)}|}{n} \text{ with n the number of days}$$

RMSE (root mean square error) is a measure of the overall error between two time series. The lower the RMSE, the better the
fit between the two series.

$$RMSE = \sqrt{\frac{\sum_{days}(\overline{F(P)} - \overline{F'(P)})^2}{n}} \text{ with n the number of days}$$





$R^2$ (coefficient of determination) is a measure of the proportion of the variance of the reference series compared to the series
to be evaluated. An $R^2$ of 1 indicates a perfect fit, while an $R^2$ of 0 indicates that there is no relationship between two series.

## Appendix F : PSC densities evolution and season changes at 50-70 hPa

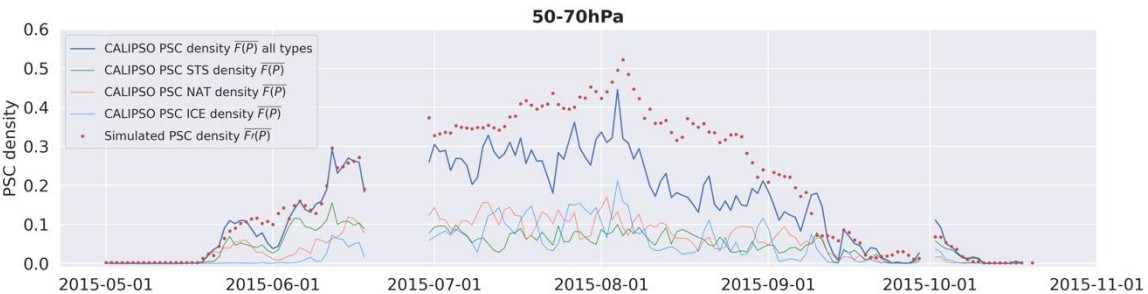

**Figure F1: Time series of daily average PSC densities $\overline{F(P)}$ observed by CALIPSO (blue curve) and PSC densities simulated by our model $\overline{F'(P)}$ (red dots) at 50-70 hPa for the year 2015. Colored lines represent the different PSC types. Green is for STS, orange**
**for NAT and blue for ICE.**

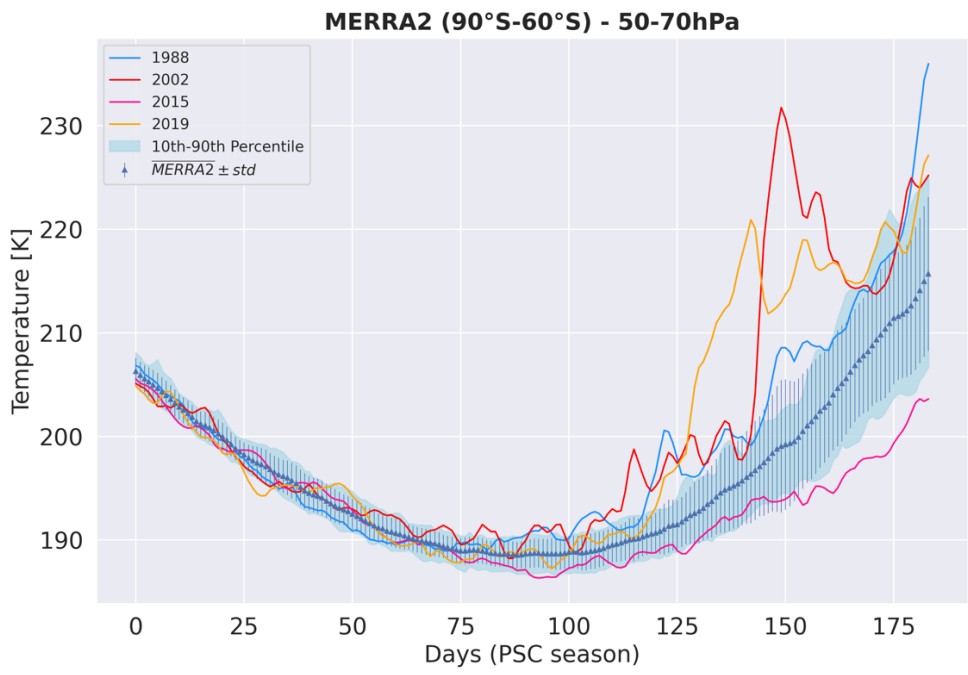



**Figure F2: Daily MERRA2 gridded stratospheric temperature average between 60°S-90°S and over each PSC season from 1980 to 2021, at 50-70 hPa. The time series goes from 1ˢᵗ May (day 0) to 31ˢᵗ October (day 184). The mean of MERRA2 (blue triangles) and**

**its standard deviation (blue vertical lines).**

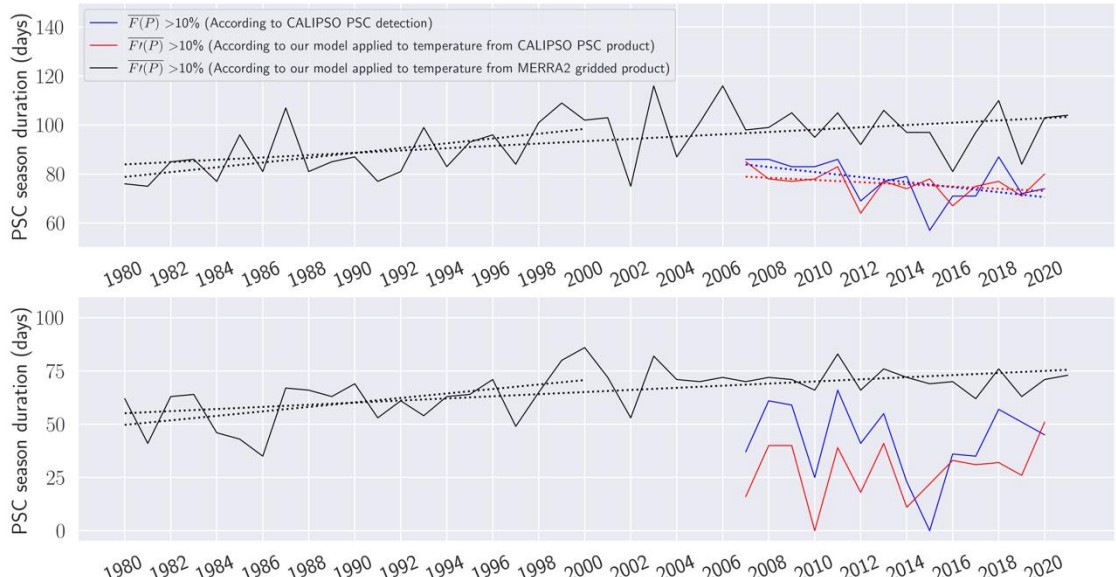

**Figure F3: The PSC season duration of P25% (a) and P40% (b) from 1980 to 2021 at 50-70 hPa. The black line represents the PSC season duration according to our model applied to gridded MERRA2 temperatures. The blue is for observations and the red for our**

**model applied to MERRA2 temperatures contained in the PSC product. Dashed lines represent statistically significant regressions.**





## Appendix G : PSC densities evolution and season changes at 100-150 hPa

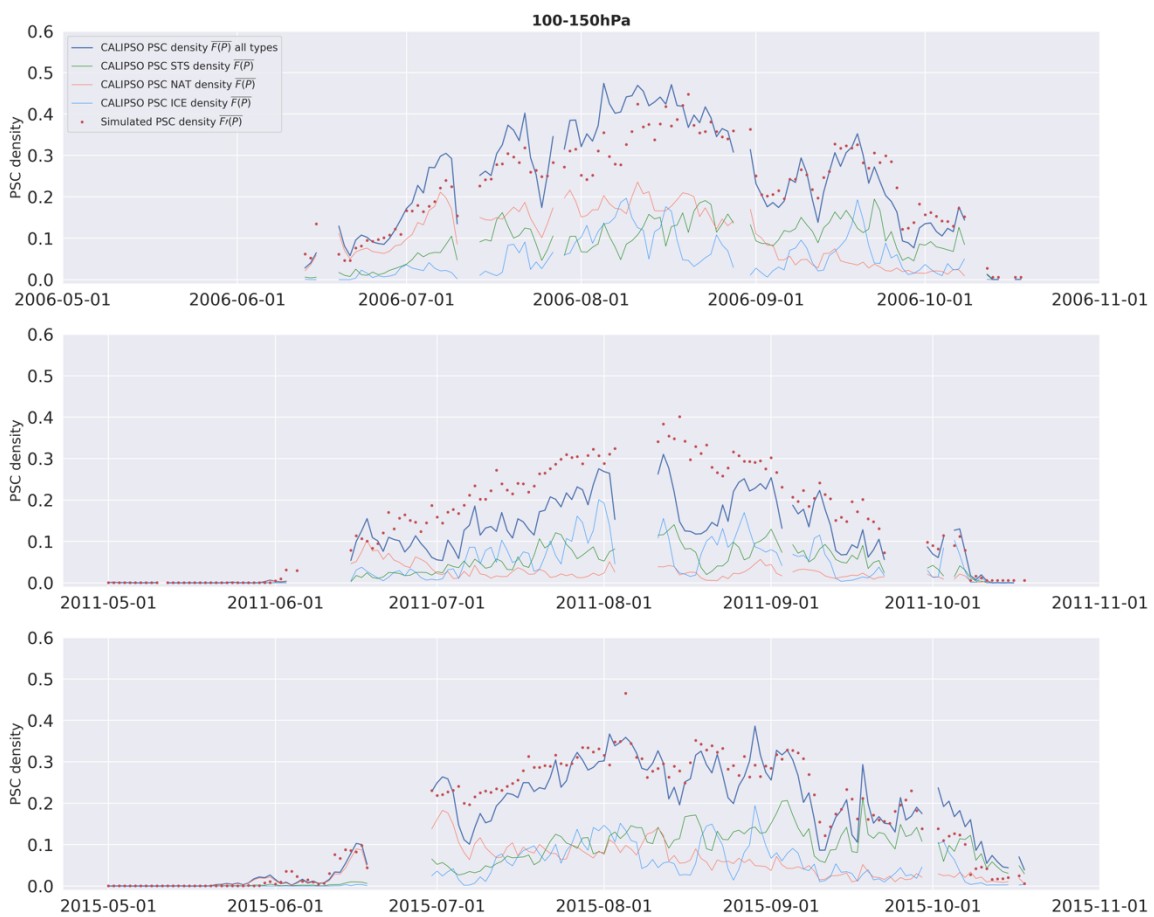

**Figure G1: Time series of daily average PSC densities $\overline{F(P)}$ observed by CALIPSO (blue curve) and PSC densities simulated by our model $\overline{F'(P)}$ (red dots) at 100-150 hPa for three years (a): 2006, (b): 2011 and (c): 2015. Colored lines represent the different PSC types. Green is for STS, orange for NAT and blue for ICE.**






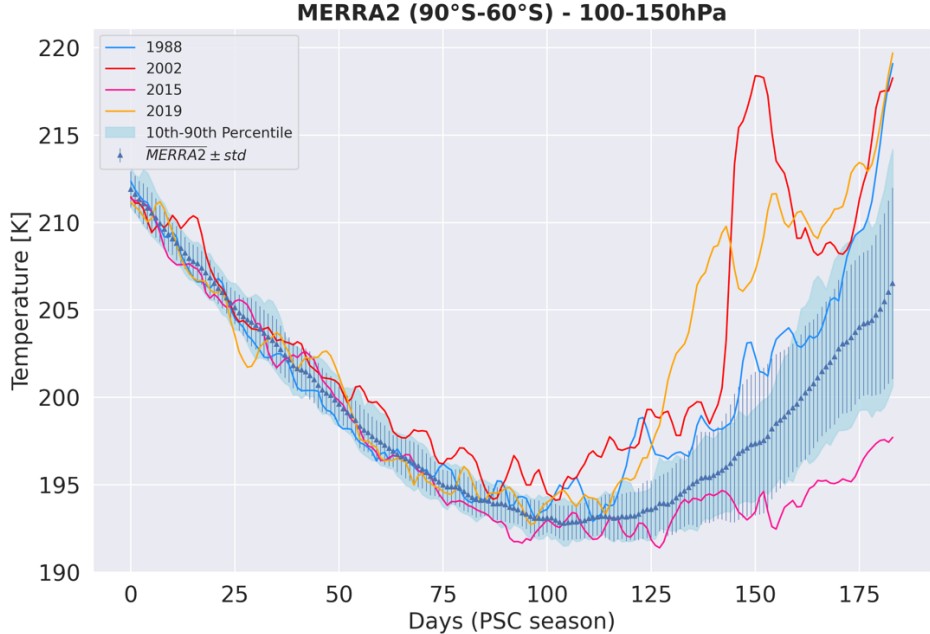

**Figure G2: Daily MERRA2 gridded stratospheric temperature average between 60°S-90°S and over each PSC season from 1980 to 2021, at 100-150 hPa. The time series goes from 1$^{st}$ May (day 0) to 31$^{st}$ October (day 184). The mean of MERRA2 (blue triangles) and its standard deviation (blue vertical lines).**

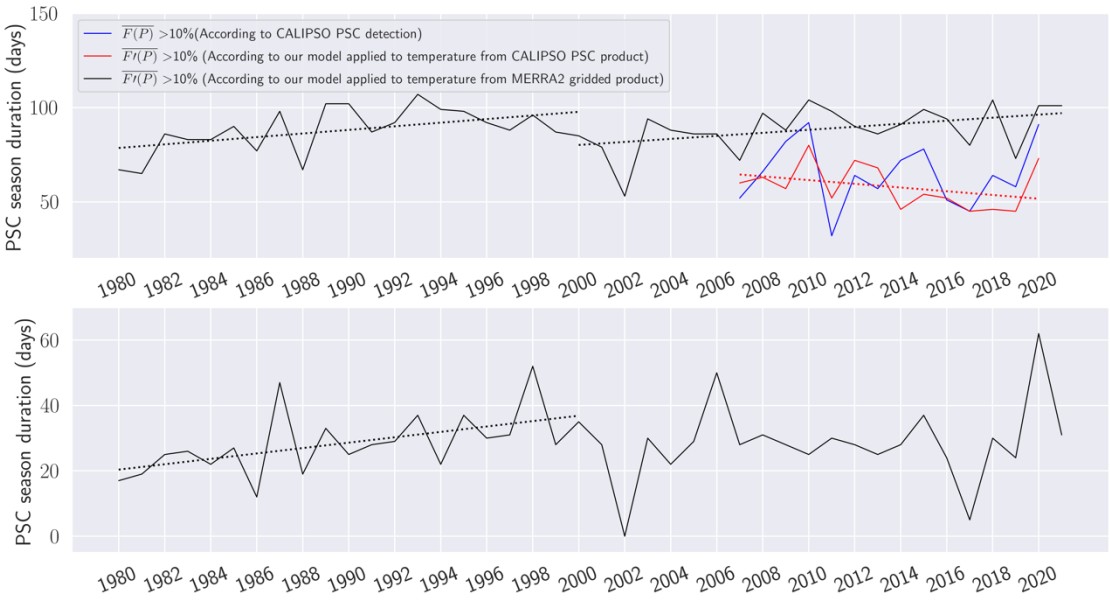

**Figure G3: The PSC season duration of P25% (a) and P40% (b) from 1980 to 2021 at 100-150 hPa. The black line represents the PSC season duration according to our model applied to gridded MERRA2 temperatures. The blue is for observations and the red**





**for our model applied to MERRA2 temperatures contained in the PSC product. Dashed lines represent statistically significant regressions.**