# Peer review of "Investigating long-term changes in polar stratospheric clouds above Antarctica during past decade: A temperature-based approach using spaceborne lidar detections"

_EGUsphere, 2024_

## Author Comment (AC1)

**Reply to Reviewer 1**

*The authors present a method for estimating Antarctic PSC cover from modelled temperatures based on a simple temperature threshold and the verification of PSC occurrence from spaceborne CALIPSO lidar measurements. The work is of interest to the readers of ACP. However, the presentation style gives the impression that a thesis was transformed into a publication without properly accounting for the rigorous trimming that is usually advisable for such a process. This is reflected in a somewhat unfocussed and repetitive presentation of findings, redundant text, the inclusion of results that don't necessarily advance the reader's understanding, and an extensive appendix. Unfortunately, the work does not include a prognosis of the likely development of future PSC occurrence. While it's debatable that this issue is implied by the title, it would certainly increase the importance of this work. Therefore, major revisions are needed to improve the quality of the paper.*

We thank the reviewer for their constructive comments which helped us to improve the quality of the paper. We have made major revisions to the article to synthesize its content, considerably reducing the length of the text, the number of steps to present results, the number of figures (from 16 to 10), and the Appendix. We have modified to title to clarify that in this work we investigate only past long-term changes in polar stratospheric clouds (see also our reply to the last comment). Below we answer to all comments point-by-point.

1.  *I suggest to omit the redundant text related to what will be presented in each section. This also holds for the abundant references to other sections and figure elements that should be described in the figure captions (red dots, solid line, etc)*

The reviewer suggests removing the introductory text at the beginning of each section, and the needless references to Figures, Sections, and figure elements. Following this comment, in the revision we have simplified the text in many places, for instance by removing references to Sections and Figures and limiting references to figure elements. We simplified the introductory text of each section that helps structure the paper.

2.  *Section 2.1 contains lots of information that isn't really relevant for this study. I suggest shortening to just some basic information on CALIPSO and the PSC mask v2.*

The reviewer suggests shortening Section 2.1. This was also mentioned by another reviewer. In the revision we have shortened Section 2.1 substantially by keeping only information relevant to the present results.

3.  *The authors need to be more precise regarding their wording. They refer to CALIPSO measurements as points. However, it is not always clear if this means profiles or*

*height bins. This is particularly important in Section 3.1 where it is unclear if the constructed grids include profiles per grid cell (N?), PSC bins per pressure level (n?), or a mixture of both. I also disagree with the term simulating PSC densities. The developed statistical model has no prognostic capabilities but rather uses a threshold to ESTIMATE the expected PSC densities.*

The reviewer points out several places where a poor choice of terms leads to lack of clarity in the paper, and suggests several word choices that would benefit from being more precise. In response to this comment, we have made efforts to select appropriate terms when required, in particular in the methodology section, and clarify the definition of the quantities used in the methodology. We hope that thanks to these efforts the revision is clearer and less affected by ambiguities. We also changed the term "simulating PSC densities" by "estimating PSC densities".

4. *The authors should elaborate on the transformation of CALIOP data from their native height resolution to that imposed by MERRA-2 pressure levels which are not equidistant. It would be interesting to learn about the number of CALIOP height bins within the different pressure levels and how this might affect PSC detection rate. Is a PSC detected as long as there's at least one PSC height bin in the PSC mask v2 product? Or is a fractional threshold used for PSC detection, i.e. a certain percentage of height bins in a layer has to feature PSCs?*

The reviewer points out a lack of clarity in our description of the regridding process through which PSC detections are transferred from the PSC Product grid (which uses equidistant CALIPSO native height bins along the vertical dimension) to the MERRA2 pressure levels (which are non-equidistant).

When we regrid the data for a given day, we first find the CALIPSO profiles that are located in a 2D gridbox (Latitude, Longitude). Then, among those profiles we identify the vertical height bins that fall into each MERRA2 pressure range, using the values of pressure by height bins present in the CALIPSO PSC product. This means that a given 3D gridbox (Latitude, Longitude, Pressure) contains a number of height bins, from a different number of separate profiles. The plot below shows as an example of a 3D gridbox that contains 9 profiles falling in the gridbox lat-lon coordinates, each profile having 8 height bins falling in the gridbox pressure range, for a total of 72 height bins. To illustrate the methodology, in this plot, 18 triangles show height bins where a PSC was present according to a fake PSC product. In each 3d gridbox such as this one we count the total number of CALIPSO height bins $N$ coming from various profiles (Fig. 1 in the paper, top left), and the number of CALIPSO height bins where a PSC was detected $n$ (Fig. 1, top right). Then we compute $F = n/N$ to obtain the PSC density for that 3D gridbox. In the example below, $F = 18/72 = 0.25$. In the revision we have updated the text to include these clarifications.

[Figure]

9 CALIPSO profiles in a 3d gridbox (Latitude, Longitude, Pressure). Blue lines represent CALIPSO profiles, points and triangles are set at each CALIPSO height bins. Red triangles represent PSC detections.

As noted by the reviewer, the number of CALIPSO height bins is not constant for each MERRA2 pressure level. This number will also change from one CALIPSO profile to the next, depending on each profile's coordinate, vertical pressure profile, and sunlight conditions (as the PSC product only considers nighttime CALIPSO measurements). The plot below shows, for one day chosen at random (2007/07/15), the mean number per profile of CALIPSO height bins within each MERRA2 pressure level when considering all profiles south of 60°S. The number of height bins within each pressure range is not constant, but there are always more than 10 height bins in a given pressure range on average. The blue region shows the standard deviation of the number of height bins by pressure level for that day. It is always 1 or smaller, showing the number of CALIPSO height bins per pressure level remains relatively constant across the south Antarctic region.

[Figure]

Average number of CALIPSO height bins within MERRA2 pressure levels per profile

5. *The presentation of the development of the statistical model in Section 3 should be improved. The current mix of presenting findings for a single day (Figure 1), a month (Figure 2), a year (Figure 4), and finally the entire time series is rather confusing and not well motivated. I suggest to include a flow chart that outlines the steps in the development of the statistical model and where iterative loops are involved. Right now, it is not entirely clear if the authors developed the model first and adapted the temperature threshold subsequently or vice versa. The text in Section 3.1 indicates that the analysis goes back and forth between CALIOP's native height spacing and the pressure levels. I suggest to stick to a fixed height grid in the presentation of results and to cover the transformation between grids in more detail in the methods section.*

The reviewer notes that the presentation of the statistical model (Section 3) is currently confusing, and suggests improvements. In the revision we made an effort of reformulation and presentation. We now explain the method using a single pressure level (50-70hPa) instead of three. We include data from a single day (Fig. 1) in order to illustrate the process of switching from CALIPSO profiles to gridded data, but otherwise all the results relate to the entire time series over the whole CALIPSO observational period (2006-2020). Following

the reviewer's suggestion, in the revision we also now include a flow chart (Figure 2), which we refer to throughout all Section 3. We have also revised the text to 1) clarify the relationship between the model and the choice of temperature threshold and 2) lift possible ambiguities between vertical grids (see also our answer to the previous comment).

> 6. *I suggest to omit Section 4. While the focus in individual years is useful to demonstrate the skill of the statistical model, the authors are going in circles by inferring fit parameters and temperature thresholds for a single year and subsequently applying it to the same year. A truly independent assessment would either need to split the available observations into data for training and verification or apply fit parameters and temperature thresholds derived for another year or the entire time series. I suggest to go with the latter, i.e, skip directly to Section 5.*

The reviewer suggests removing Section 4 and presenting directly the results about the CALIPSO observational period. Following this comment we removed Section 4 and now directly dive into results over the entire CALIPSO observational period.

> 7. *The discussion in Section 5 nicely presents the findings and the capability of the statistical model. However, the presentation is rather repetitive going from one pressure level to the next. I suggest to further condense the results into a presentation that covers all height levels (as in Figure 12) and years (as in Figure 7) with PSC density variations as colour map. The same applies to Section 6. Another thing that remains entirely unclear to me are the PSC density thresholds of 10%, 25%, and 40%. How have these values been derived? Should they be the same at all height level? I suggest to add some text that motivates their selection.*

The reviewer notes that the presentation choices in Sections 5 and 6 make them rather repetitive, as each time we dealt with the three pressure levels in succession. The reviewer suggests condensing the result in one figure. Following this comment, the plot below shows the composite PSC density variations according to CALIPSO observations (top) and PSC density variations estimated by our model (bottom) from May to late October, using the complete CALIPSO observational period. However, those figures do not allow us to easily include the standard deviations or the maximas and minimas that were included in Figure 6b, 8b, and 10b. We could include those values in additional figures, but that would make comparisons harder and somewhat defeat the purpose (i.e. simplify by limiting the number of figures). Searching for a middle ground, in the revision a single figure summarizes PSC densities (observed and evaluated from temperatures) at all pressure levels. The figure is certainly imposing but allows everything to be shown. Instead of going back and forth between pressure levels, now the discussion addresses all pressure levels in the same section.

[Figure]

Regarding the PSC density thresholds of 10%, 25%, 40%, we thank the reviewer for pointing out the fact that these values should not be the same for all pressure levels, since the PSC densities are not the same over each pressure level (at 10-20hPa densities are for instance particularly low). Following this comment, we have revised our methodology to identify the beginning and end of the PSC season. Instead of using a fixed threshold of 10%, we now calculate, for each pressure level, the standard deviation of the PSC densities averaged over 2006-2020 (blue line, Figure 6b). As a consequence, in the revision P10% has become Pσ. The P25% and P40% thresholds were removed, and in the revision we now highlight periods when the PSC densities are large using a P2σ threshold (twice the standard deviation). We have updated the discussion to take into account the new results. Since our model estimates well the PSC densities, changing thresholds does not affect the accuracy of our model concerning the start and end dates of the PSC season. Compared to our previous results, slightly higher thresholds mean PSC seasons generally start later and end earlier between 20 and 150 hPa. The exception is the 10-20hPa level where thanks to a lower threshold (0.04) the PSC season starts earlier and ends later. By consequence, the PSC season is now longer at 10-20 hPa and shorter on pressure levels below. Changes in PSC season duration (Fig 9, previously Fig. 16) are not impacted by these new thresholds.

8. *Section 6 presents the application of the inferred statistical model to a time period not covered by the CALIPSO time series. This is the core advancement of the work as it expands available knowledge to climate time scales. However, by including all model data south of 60 degree S, the authors have skipped a step that would have allowed for a fairer assessment of the long-term development of PSC densities. Why not consider the same region covered by CALIOP observations first and extending*

*southward afterwards? This might even provide you with a quantitative explanation or a correction for the discrepancies in Figures F3 and G3. Simply accounting this to the difference in area without having a closer look doesn't seem right to me. In that context, I would also have expected a more conclusive discussion (with references) regarding the step in the trend line at around 1999 in Figures 13 and 15. Is there a physical explanation for those? There must be time series of stratospheric temperature or large-scale circulation that could be considered? If there have been events such as SSW, it's worthwhile to mark them in the plots of time series. Also relating to the trends: the authors refer to a significant increase. If they have performed significance tests, I suggest to provide the results (e.g., kind of test, p-value) to support their statements.*

The reviewer suggests that when considering gridded temperatures, contrast results over the 60°S-82°S region with the complete polar region. This was also mentioned by the other reviewer. We thank the reviewers for correctly pointing out that we had all the required data to explore this hypothesis. The new results obtained by following these suggestions are included in what is now Section 5 (previously section 6). We now discuss what these results reveal about how representative CALIPSO PSC observations are of the more general PSC season.

Concerning the trends found in the pre-CALIPSO period, there is indeed a discontinuity in the MERRA2 stratospheric temperatures around 1999 due to the assimilation of new measurements. We now discuss these issues in the text. The significance of the trend was assessed by testing whether the slope coefficient of the regression model was statistically different from zero. This was done using a hypothesis test, where the null hypothesis was that the slope coefficient equals zero, indicating no trend over time. We used a significance level (alpha) of 0.05, which is a common threshold in statistical analysis. This information is now explained in the text. We provide a reference for the statistical recipes that were used to evaluate significance.

We now indicate on time series plots the years characterized by SSW and volcanic eruptions (Fig. 10 now part of section 6).

9. *I strongly advise the authors to add a section in which the statistical model is applied to the output of a climate model to estimate PSC coverage and season length until 2100. This would add a major novelty to the study and strengthen its overall scope. I can understand that the authors might want to focus on this during subsequent work. However, using a single climate model that best resembles stratospheric temperature of MERRA-2 during 1980 to 2020 to demonstrate the feasibility of their approach wouldn't impede more detailed follow-up studies that might involve a set of climate models.*

The reviewer strongly suggests adding a Section in which our model is applied to temperatures from at least one climate model in a given emission scenario to estimate PSC densities and the evolution of the PSC season duration over the next century.

We understand the reasons why the reviewer strongly advises for including these additional results, and we seriously considered following this path during the initial writing phase of the paper. We eventually decided against going this route in the article under review for several reasons:

- Due to substantial differences in stratospheric temperatures between reanalyses and GCM output, applying our model to output from GCMs requires modifications to the methodology, including a recalibration of the temperature vs PSC density regressions (as we note in the conclusion), to fit the observed PSC cover to the stratospheric temperatures generated by the GCMs. Preliminary tests have shown that this is not as straightforward as one could think, as differences in spatial resolution (for instance) have to be accounted for. Our tests have also shown that variations of polar stratospheric temperatures on various timescales often adopt different behaviors in GCMs and reanalyses. These differences might require the development of adaptations to our methodology. Thus even considering temperatures from a single GCM would require a substantial expansion of the methodology, adding structural complexity and length to the paper.

- To keep the scope of the paper in check would require considering output from a single GCM, as noted by the reviewer. By doing so, however, we would not be able to provide useful context, as in: are the predictions from this GCM representative of all GCMs? Are they close to the upper or lower bounds? Is a specificity from the considered GCM affecting the results in one way or another? Not being able to consider these questions would seriously constrain the discussion of results and limit their usefulness. Thus it is not clear to us what would be gained by following this path, apart from being able to show the method works. Showing PSC previsions over the next century without being able to interpret them would without doubt prove frustrating to readers.

- To make such efforts useful, we think it is necessary to include comparisons of predictions from several GCMs, perhaps in several emission scenarios, and discuss in detail specificities of each, and how they might relate to the predictions. Doing so in a rigorous manner would require substantially expanding the scope and the length of the paper, which was already too long according to the first round of reviews.

For these reasons, we think that it is more reasonable to address the aspect advised by the reviewer in a non-superficial way in a dedicated paper. We hope to present the results expected by the reviewer in an upcoming article.

---

## Author Comment (AC2)

**Reply to Reviewer 2**

*In this work, the authors present a statistical model to evaluate the existence period of Polar Stratospheric Clouds (PSCs) from global gridded stratospheric temperature datasets. The model parameterization is derived from PSC-observations performed by the CALIOP lidar on the CALIPSO satellite between 2006 and 2020. Subsequently this model is used to analyse the trend of the PSC season length over Antarctica at different stratospheric pressure levels over an extended period from 1980 to 2020 based on the MERRA2 reanalysis dataset.*

*In general, the manuscript is well written albeit a bit lengthy in some places where the material could be presented more compressed with less repetitions. I would suggest to really concentrate only on the really necessary parts and move some of the less significant items in the appendix. Content wise the manuscript fits into the scope of ACP and I would support its publication after consideration/implementation of the following comments.*

We thank the reviewers for their constructive comments which helped us to improve the quality of the paper. Following these comments and similar ones from the other reviewer, we made an effort in the revision to clarify and summarize the repetitive and long Sections. We have considerably reduced the length of the text, the number of steps to present results, the number of figures (from 16 to 10), and the Appendix. We answer below point-by-point to all comments.

- *Chapter 2.1:*

  *Seems too long with information not needed in the following. Please concentrate on describing the dataset which is used within the study and put references for further reading.*

The reviewer suggests shortening Section 2.1. Following this comment and a similar one from the other reviewer, we have shortened Section 2.1 by trying to keep only information relevant to the paper.

- *Chapter 2.2:*

  *Here (or later in the discussion section) a discussion on the reliability of MERRA2 would be valuable to be able to judge on the conclusions drawn on the pre-CALIPSO periods. As an example, one may refer to the "SPARC Reanalysis Intercomparison Project (S-RIP) Final Report" or other work, especially comparing temperatures with ERA-5.*

We thank the reviewer for this extremely useful SPARC reference. Based on its contents, we added a discussion about the reliability of MERRA2 temperatures in the polar lower stratosphere in Sect 2.2 and Sect. 5 to discuss the significance of trends found over different timescales.

- **193: 'fits the best for most of the plots is a polynomial of degree 2'**

  *What is the criterion for the statement 'best'? A higher-order polynomial would mathematically fit better but there might be other reasons (e.g. simplicity) to choose 2$^{nd}$ order. What is the norm which is minimized for the fit (RMS)?*

We have performed tests with polynomials of different orders and found that degree 2 is the smallest order giving good performance (ie small determination coefficients). We have modified the text to explain that. Our python code uses from numpy the polyfit function, which indeed minimizes the squared error as the reviewer suggests.

- **218: 'The $Tpsc$ and parameters which lead to the smallest MAE are selected for the month and pressure level considered'**

  *How well defined is this minimum? A plot of MAE as f(Tpsc) would be helpful to judge on the uniqueness of the result. Further, the choice to use monthly changing threshold temperatures seems a bit arbitrary. Have you tried to perform some kind of 'smooth' transition between the monthly Tpsc values or what is your argument to use such a 'coarse' binning.*

In our study, we calculated the MAE for each Tpsc and chose the Tpsc which gives the smallest MAE. This is now clarified in the text. To investigate how well the minimum is defined, the plot below presents for each pressure level in June the MAE as a function of Tpsc. Minimas appear well-defined enough to specify Tpsc with a precision of 1-2 Kelvin.

[Figure]

Mean absolute error (MAE) between observed and evaluated PSC densities depending on the temperature threshold TPSC at different pressure levels

Defining Tpsc using monthly timescales makes it possible to take into account changes in chemical species available for the formation of PSCs. As a given Tpsc provides a set of (a,b,c) coefficients from the polynomial regressions which estimate the PSC densities, we cannot interpolate the Tpsc (since we cannot interpolate the regression coefficients). Our tests have shown it is possible to calculate threshold temperatures on sub-monthly scales (for instance 15 days), this does not change the results significantly. Shorter timescales do not provide enough data points to the regressions.

- *238: 'The difference is particularly important at lower altitudes'*

  *This finding should be supported by references to previous publications.*

- *248: 'there is often a large difference between the temperature threshold $T_{NAT}$ at'*

  *Can you support this statement by pointing to papers describing these obvious differences based on CALIOP observations?*

We address these two comments as one as they relate to the same issue. The noted sentences (lines 238 and 248) describe Tpsc in Figure 3. Tpsc is a temperature threshold that we defined in the paper and is required to apply our methodology. It is therefore not referenced in existing literature, and we cannot rely on the literature to explain the differences between TNAT and Tpsc. We do not expect Tpsc to be necessarily equivalent to TNAT. We apologize for the confusion, and tried to clarify the text to avoid misunderstandings.

- ***Chapter 4:***

  *Here only the simulated PSC densities on basis of monthly temperature thresholds for the related year are shown. I think the results obtained with the multi-year derived T-thresholds are much more relevant. Thus, I would strongly suggest to show and discuss also those. In principle this is shown in Figs. 6-8 in chapter 5, albeit so squeezed that the curves cannot be distinguished.*

Following this suggestion and a similar one from the other reviewer, the original section 4 (2009 and 2010 PSC seasons) was removed. Consistent with this comment, the new section 4 now focuses on the results obtained with the multi-year derived T-thresholds. In order to facilitate the reading of the squeezed Figures 8a, Appendix C now describes for the pressure level 50-70 hPa, each PSC densities observed and estimated by our model for each year from 2007 to 2020.

- ***551: 'This longer period can be attributed to the gridded dataset including latitudes south of 82°S, absent from the CALIPSO dataset.'***

  *This statement can easily (and should) be substantiated by performing the respective analysis only down to 82S.*

We thank the reviewer for this appropriate suggestion, also noted by the other reviewer. We updated what is now Section 5 (previously Section 6) by presenting results restricted to the 60°S-82°S region. The new results confirm that the considerably longer PSC seasons are indeed due to the inclusion of the region poleward 82°S.

- ***Chapter 6:***

  *These findings strongly depend on the accuracy of the MERRA2 analysis, esp. on their trends. Please discuss the influence of related uncertainties on your findings and, if possible, point to supporting material.*

Following this comment and a similar one on Section 2.2, Section 5 now includes a discussion of the accuracy and biases of MERRA2 stratospheric temperatures and their relation to the trends we found in the PSC season duration over the 1980-2021 period.

- ***Data availability :***

  *The data used for the model calculations (e.g. Tpsc, polynomial coefficients) have to be made publicly available in digital form.*

Following this comment, a CSV file (data_statistical_model.txt) describing for each pressure level and month the Tpsc and associated regression coefficients has been uploaded as supplementary data on ACP.

---

## Editor Decision (ED1)

[revised manuscript text omitted]
 over 60° S-82° S (black) and over 60° S-90° S (green). In blue the PSC season duration observed by CALIPSO and in red estimated by our model applied to temperatures contained in the CALIPSO PSC product. The red points represent the volcanic eruptions and purple stars the SSW. Only statistically significant trends are shown in dashed lines.**

**Appendix E: PSC densities evolution and season changes at 50-70 hPa**

[Figure]

**Figure E1: Time series of daily average PSC densities $\overline{F_d(P)}$ observed by CALIPSO (blue curve) and PSC densities estimated by our model $\overline{F_d{}'(P)}$ (red dots) at 50-70 hPa (a) and at 100-150 hPa (b) for the year 2015. Colored lines represent the different PSC types. Green is for STS, orange for NAT and blue for ICE.**

[Figure]

**Figure E2: Daily MERRA2 gridded stratospheric temperature average between 60° S-90° S and over each PSC season from 1980 to 2021, at 50-70 hPa (a) and at 100-150 hPa (b). The time series goes from 1st May (day 0) to 31st October (day 184). The mean of MERRA2 (blue triangles) and its standard deviation (blue vertical lines).**

785